# NEXT-SCALE AUTOREGRESSIVE FORECASTING FOR TIME SERIES VIA MODULAR MULTI-SCALE DECOUPLING

## ABSTRACT

Time series forecasting underpins critical applications in finance, energy, healthcare, and transportation. Although deep models have achieved strong results, most adopt single-scale modeling or restrict multiscale processing to the input side, causing a misalignment between multiscale inputs and single-scale outputs and limiting predictive power. We introduce the **Modular Scale-wise Autoregressive Framework (MSAR)**, a model-agnostic design that forecasts progressively across multiple temporal resolutions. MSAR offers three advantages: (1) **scale-wise aligned modeling**, which disentangles heterogeneous temporal patterns by aligning inputs and outputs at each scale; (2) **scale-wise autoregression**, where coarse-scale predictions guide finer-scale forecasting through hierarchical information flow; and (3) a **modular architecture**, enabling seamless integration with diverse backbones such as CNNs, MLPs, and Transformers. Extensive experiments across a broad set of datasets and forecasting models demonstrate that MSAR achieves consistent improvements in both accuracy and inference efficiency, validating the effectiveness of scale-aligned autoregression for multiscale time series forecasting. All resources needed to reproduce our work are available: https://anonymous.4open.science/r/MSAR-E8EB.

## 1 INTRODUCTION

Time series forecasting is critical in domains such as finance, energy, healthcare, and transportation. Recent deep models—including RNNs (Salinas et al., 2020; Bergsma et al., 2023b;a; Lin et al., 2023), CNNs (Wu et al., 2022; Wang et al., 2023), MLPs (Zhang et al., 2022; Zeng et al., 2023; Yu et al., 2024; Tang & Zhang, 2025), and Transformers (Nie et al., 2023; Liu et al., 2024a; Brigato et al., 2025)—have shown strong performance. However, they typically rely on single-scale modeling, processing historical data at a fixed temporal scale. This overlooks a key property of real-world time series: temporal patterns vary across scales (Wang et al., 2024a). Forcing heterogeneous patterns into a unified representation often causes scale interference, leading to poor generalization and degraded forecasting accuracy.

To address the limitations of single-scale modeling, prior works (Geva, 1998; Guo et al., 2023; Zhang & Yan, 2023; Wang et al., 2024a; Shabani et al., 2023; Chen et al., 2024b; Wang et al., 2025; Han et al., 2025; Chen et al., 2025) introduced multiscale-mixing in input modeling to capture patterns at different scales. However, they still predict future values at a single (fine) scale, leading to a mismatch between multiscale inputs and single-scale outputs. Similarly, multi-resolution decoders (Challu et al., 2023; Kraus et al., 2024) generate coarse-to-fine intermediate components, but these are ultimately *interpolated* into a single-resolution prediction with no per-scale supervision. This misalignment complicates the learning process and weakens the model's ability to leverage multiscale information effectively, often resulting in suboptimal forecasts.

Building upon the aforementioned limitations, we propose the Modular Scale-wise Autoregressive Framework (MSAR). As shown in Figure 1, MSAR performs **scale-aligned** and **decoupled modeling**, where each scale is modeled independently. This explicit alignment helps isolate heterogeneous temporal patterns and simplifies the learning process. Motivated by Tian et al. (2024), we introduce a **scale-wise autoregressive** forecasting strategy. This structured design clearly distinguishes MSAR

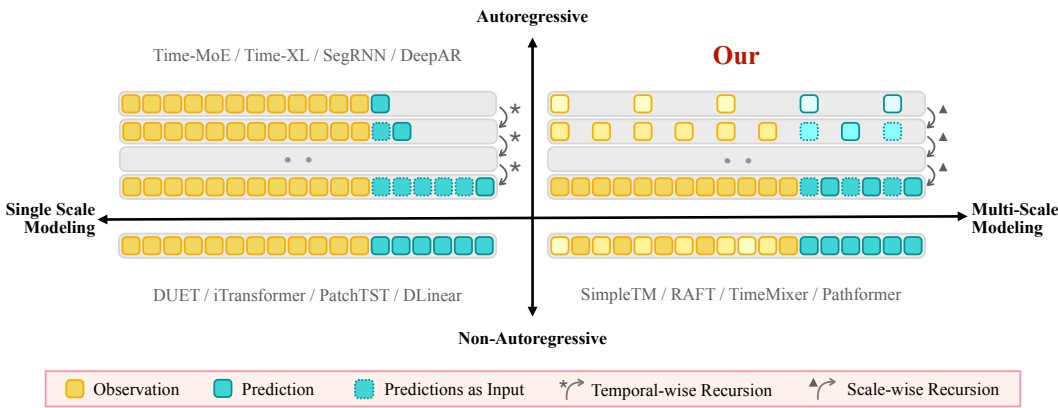

Figure 1: Comparison of forecasting paradigms. MSAR introduces scale-wise autoregression with aligned multi-scale inputs and outputs. Coarse-scale predictions serve as contextual guidance for finer-scale forecasting through hierarchical information flow.

from prior approaches that rely on single-scale decoding (Shi et al., 2025; Liu et al., 2024b; Qiu et al., 2025; Liu et al., 2024a; Nie et al., 2023; Zeng et al., 2023), from cross-scale mixing without resolution consistency (Chen et al., 2025; Han et al., 2025; Wang et al., 2024a; Chen et al., 2024b), and from multiresolution decoders (Challu et al., 2023; Kraus et al., 2024). In contrast, MSAR enforces **explicit per-scale supervision** and **strict within-scale alignment**, enabling principled hierarchical autoregressive refinement across scales.

Specifically, MSAR adopts a scale-wise modular framework that enables flexible implementation across different forecasting models. This design makes the MSAR **model-agnostic** and easy to integrate into a variety of backbone architectures. To coordinate the prediction process across scales, MSAR introduces a simple yet effective information flow mechanism, where predictions from coarser scales serve as contextual signals to guide finer-scale forecasting. Furthermore, to mitigate potential errors propagated from earlier predictions, we incorporate a lightweight refinement module that selectively fuses coarse predictions with current-scale inputs. This refinement process improves prediction accuracy and enhances consistency across scales.

MSAR offers three key advantages over prior approaches:

❶ Scale-wise aligned modeling: Unlike previous methods that apply multiscale-mixing (Chen et al., 2025; Han et al., 2025; Wang et al., 2024a; Chen et al., 2024b) only to input features, MSAR performs fully separated modeling at each scale, with inputs and outputs aligned within the same scale. This design reduces learning complexity by isolating heterogeneous temporal patterns and avoiding interference across scales.

❷ Scale-wise autoregression: MSAR adopts a scale-wise autoregressive strategy (Tian et al., 2024), where predictions from coarser scales serve as contextual signals for finer-scale forecasting. This hierarchical structure improves the model's ability to capture long-term dependencies while enabling progressive refinement of short-term patterns, thereby enhancing both accuracy and interpretability.

❸ Modular and model-agnostic design: Each scale in MSAR is processed by an independent forecasting module without architectural constraints. This modular design enables seamless integration with diverse backbone models—such as CNNs (Wu et al., 2022), MLPs (Zeng et al., 2023), and Transformers (Nie et al., 2023; Liu et al., 2024a), and allows flexible adaptation to varying input lengths.

## 2 RELATED WORK

### 2.1 TIME SERIES FORECASTING

**Single-Scale Modeling.** Modern time series forecasting is predominantly driven by deep learning models, including RNN-based (Lin et al., 2023), CNN-based (Wu et al., 2022), Transformer-

based (Nie et al., 2023; Liu et al., 2024a), and MLP-based (Zeng et al., 2023; Qiu et al., 2025) architectures. However, these methods typically operate at a single temporal scale and fail to capture distinct patterns that emerge across different resolutions. As a result, long-context inputs often lead to overfitting, as the model attempts to fit both global trends and local fluctuations within a unified representation, without scale-aware separation (Liu et al., 2024b).

**Multiscale Modeling.** To capture temporal dynamics across different frequencies, several recent works have introduced multiscale modeling into time series forecasting. SimpleTM (Chen et al., 2025) employs wavelet-based tokenization to generate scale-specific representations for Transformers. RAFT (Han et al., 2025) leverages retrieval across different periodicities to inject external multi-scale patterns into forecasting. TimeMixer (Wang et al., 2024a) employs grouped mixing layers to model dependencies across temporal resolutions in a unified encoder. Pathformer (Chen et al., 2024b) constructs adaptive pathways to route information across temporal resolutions and distances. These methods effectively encode multiscale patterns in the *input sequence* but still predict at a *single temporal scale*, creating a mismatch between input and output resolutions. In contrast, our MSAR performs *scale-aligned*, coarse-to-fine *autoregressive* decoding: each stage only consumes inputs at its assigned resolution and is causally conditioned on the previous (coarser) stage, enabling consistent multiscale modeling across both input and output without fusion-based mismatch.

**Multi-Resolution Decoders.** Many recent architectures employ *multi-resolution decoders*, where intermediate temporal scales are generated or refined before producing the final forecast. Crossformer (Zhang & Yan, 2023) and MR-Transformer (Zhu et al., 2023) construct hierarchical or segment-level multi-resolution pathways, yet only the *final fine-scale output* is supervised. Scaleformer (Shabani et al., 2023) performs iterative multi-scale decoding, but all intermediate resolutions are fused into a single prediction through pooling or learned interpolation. xLSTM-Mixer (Kraus et al., 2024) mixes multi-span temporal memories inside a multi-resolution decoding stack but still emits a single-resolution forecast. N-HiTS (Challu et al., 2023) explicitly reconstructs intermediate blocks at multiple resolutions, but these are subsequently upsampled and aggregated rather than decoded as separate supervised outputs. Thus, although these methods use multi-resolution *decoding mechanisms*, none of them performs *explicit per-scale decoding with per-scale targets*.

### 2.2 AUTOREGRESSIVE APPROACHES FOR SEQUENCE MODELING

Autoregressive (AR) modeling has been a widely adopted paradigm in time series forecasting. DeepAR (Salinas et al., 2020) and SutraNets (Bergsma et al., 2023a) generate the future sequence one time step at a time, where each prediction depends on historical inputs and previously generated values. SegRNN (Lin et al., 2023) extends conventional RNNs with segment-wise recurrence and parallel multi-step decoding. C2FAR (Bergsma et al., 2023b) adopts a hierarchical binning strategy to generate each value autoregressively from coarse to fine levels of numerical precision. Despite their differences, all these methods operate at a *single temporal scale*, which limits their ability to capture complex temporal dependencies that span across multiple resolutions. Recent time-series foundation models such as Time-MoE (Shi et al., 2025) and Time-XL (Liu et al., 2024b) also adopt autoregression over time steps, leveraging large model capacity and extensive pretraining to achieve strong performance. However, their focus is on scaling parameters and training data, whereas our work centers on paradigm improvements.

Motivated by VAR (Tian et al., 2024), we propose the MSAR, which formulates forecasting as a *next-scale prediction* task across scale-aligned temporal resolutions. MSAR takes as input a historical time series and decomposes both the inputs and prediction targets into multiple aligned temporal scales. Each scale is modeled separately using a plug-in forecasting module, and predictions from coarser scales are progressively used to condition finer-scale forecasts, enabling a hierarchical flow of information.

## 3 METHODOLOGY

**Problem Formulation** Let the time series data be represented as: $\mathbf{X}_{1:T} = \{\mathbf{x}_1, \mathbf{x}_2, \ldots, \mathbf{x}_T\} \in \mathbb{R}^{T \times C}$, where $T$ denotes the length of the look-back window and $C$ represents the number of input

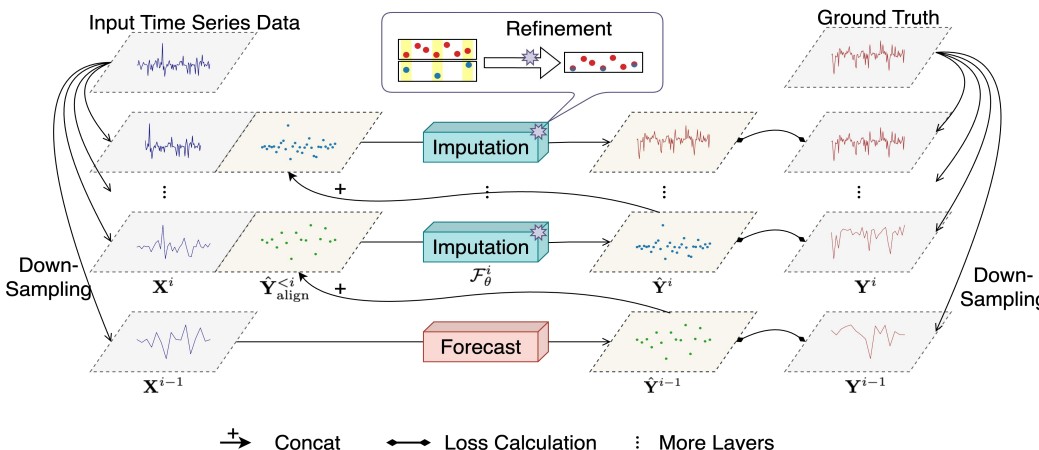

Figure 2: Overview of the MSAR framework. Conceptual illustration of multi-scale input/output modeling and the progressive imputation mechanism.

channels. The goal is to predict the future sequence: $\mathbf{Y}_{1:H} \in \mathbb{R}^{H \times C}$, where $H$ is the forecast horizon. For simplicity, we directly use $\mathbf{X}$ and $\mathbf{Y}$ to represent the input and output data, respectively.

**Notations** We define $\mathbf{X}_{1:T}^{\Delta} \in \mathbb{R}^{\lfloor \frac{T}{\Delta} \rfloor \times C}$ as the downsampled sequence over time steps, where $\lfloor \frac{T}{\Delta} \rfloor$ represents the dimension after downsampling. Similarly, we define $\mathbf{Y}_{1:H}^{\Delta} \in \mathbb{R}^{\lfloor \frac{H}{\Delta} \rfloor \times C}$ as the prediction results with a time interval of $\Delta$.

### 3.1 TECHNICAL MOTIVATION

To understand how scale alignment and temporal granularity affect forecasting performance, we conduct experiments on the ETTh1 dataset with a fixed prediction horizon of $H = 336$. As shown in Table 1, we evaluate models under three input settings: (1) full-resolution input ($T = 576$); (2) downsampled [1] input with full-resolution output ($T = 576^{\Delta=6}, H = 336$); and (3) fully scale-aligned input-output ($T = 576^{\Delta=6}, H = 336^{\Delta=6}$). In addition to standard MSE, we report $\text{MSE}^{\Delta=6}$, which measures prediction quality under 6× temporal downsampling.

We highlight three key observations:

**(1) Scale misalignment degrades performance.** When input and output scales are not aligned (e.g., $T = 576^{\Delta=6}, H = 336$), performance consistently drops compared to the fully aligned setting. This degradation occurs despite the downsampled input offering more context in real time steps, underscoring the importance of maintaining resolution consistency between input and output.

Table 1: Forecasting results on the **ETTh1** dataset. $MSE^{\Delta=6}$ reflects the MSE under 6× downsampling. $576^{\Delta=6}$ and $336^{\Delta=6}$ correspond to context and prediction sequences with 6× downsampling.

| Model | T | H | MSE | MSE$^{\Delta=6}$ |
|---|---|---|---|---|
| PatchTST | 576 | 336 | 0.480 | 0.477 |
| | $576^{\Delta=6}$ | 336 | 0.504 | 0.445 |
| | 576 | $336^{\Delta=6}$ | - | 0.564 |
| | $576^{\Delta=6}$ | $336^{\Delta=6}$ | - | **0.433** |
| DLinear | 576 | 336 | 0.445 | 0.449 |
| | $576^{\Delta=6}$ | 336 | 0.465 | 0.438 |
| | 576 | $336^{\Delta=6}$ | - | 0.457 |
| | $576^{\Delta=6}$ | $336^{\Delta=6}$ | - | **0.425** |
| TimeMixer | 576 | 336 | **0.432** | 0.435 |
| | 576 | $336^{\Delta=6}$ | - | 0.461 |

**(2) Sparse, scale-aligned forecasting improves coarse-scale accuracy.** Models trained and evaluated under the same coarse resolution ($T = 576^{\Delta=6}, H = 336^{\Delta=6}$) achieve the lowest MSE$^{\Delta=6}$,

---

[1]This operation can be implemented in PyTorch-style code as `x[:, ::Delta, :]`, where `x` is the input sequence with dimensions [`batch_size, seq_len, num_of_channels`].

demonstrating that explicitly modeling forecasting at sparse temporal scales yields more accurate coarse-level predictions—even with significantly fewer input tokens.

**(3) Multiscale-mixing improves overall accuracy but lacks scale-specific precision.** TimeMixer (Wang et al., 2024a), which adopts a multiscale-mixing strategy through grouped temporal operations, achieves the lowest overall MSE. However, its coarse-scale performance (as indicated by $\text{MSE}^{\Delta=6}$) remains inferior to that of models trained explicitly at coarse resolutions. This suggests that while multiscale-mixing enhances global modeling capacity, it may obscure scale-specific patterns and reduce fidelity at individual temporal granularities.

These findings motivate our proposed framework, which models forecasting at multiple aligned temporal scales, aiming to preserve both temporal coherence and resolution-specific expressiveness.

## 3.2 MODULAR SCALE-WISE FORECASTING FRAMEWORK

Real-world time series often exhibit temporal patterns at multiple resolutions—long-term trends unfold gradually, while short-term fluctuations vary rapidly. Modeling such heterogeneous dynamics using a fixed-resolution representation often leads to underfitting or overfitting. To address this, we propose the Modular Scale-wise Forecasting Framework (MSAR), a model-agnostic pipeline that decomposes forecasting into multiple scale-aligned subproblems. Each scale is handled independently by a dedicated forecasting module, while a lightweight information flow mechanism propagates relevant context from coarse to fine resolutions.

### 3.2.1 SCALE-WISE DECOUPLING AND PROGRESSIVE FORECASTING

We define a sequence of temporal scales with downsampling factors $\{d_1, d_2, \ldots, d_N\}$, where $d_1 > d_2 > \cdots > d_N = 1$. Given an input sequence $\mathbf{X}_{T-W:T} \in \mathbb{R}^{W \times C}$ and a forecasting horizon $H$, the scale-$i$ input and target are

$$\mathbf{X}^i = \mathbf{X}^{\Delta=d_i}_{T-W_i:T} \in \mathbb{R}^{\frac{W_i}{d_i} \times C}, \qquad \mathbf{Y}^i = \mathbf{Y}^{\Delta=d_i}_{1:H} \in \mathbb{R}^{\frac{H}{d_i} \times C}, \tag{1}$$

where $\mathbf{X}^{\Delta=d}$ denotes a deterministic stride-based sampling operator that retains every $d$-th timestamp without interpolation or filtering, thereby preserving the original temporal structure at a coarser resolution. The $W_i$ is the look-back window at scale $i$.

**Forecasting Objective.** MSAR formulates multiscale forecasting as a structured autoregressive process:

$$p(\mathbf{Y}^1, \ldots, \mathbf{Y}^N \mid \mathbf{X}^1, \ldots, \mathbf{X}^N) = \prod_{i=1}^{N} p(\mathbf{Y}^i \mid \mathbf{X}^i, \hat{\mathbf{Y}}^{<i}_{\text{align}}), \tag{2}$$

where $\hat{\mathbf{Y}}^{<i}_{\text{align}}$ aggregates coarse-scale predictions whose time indices coincide with those of scale $i$:

$$\hat{\mathbf{Y}}^{<i}_{\text{align}} = \bigcup_{j<i} \left\{ \hat{\mathbf{Y}}^j_t \,\middle|\, t \in \boldsymbol{\tau}^j \cap \boldsymbol{\tau}^i \right\} \in \mathbb{R}^{\frac{H}{d_i} \times C}, \tag{3}$$

with $\boldsymbol{\tau}^j$ the index set at scale $j$. For time points in $\boldsymbol{\tau}^i$ that receive no coarse-scale coverage, we insert zero vectors to match the target size of $\mathbf{Y}^i$:

$$t \notin \bigcup_{j<i} \boldsymbol{\tau}^j \quad \Rightarrow \quad \hat{\mathbf{Y}}^{<i}_{\text{align}}[t] = \mathbf{0}.$$

**Information Flow Mechanism.** The role of each prediction module $\mathcal{F}^i_\theta$ differs by scale. At the coarsest level ($i=0$), $\mathcal{F}^0_\theta$ acts as a *forecasting model*, generating an initial coarse-resolution prediction solely from the input $\mathbf{X}^0$. For higher scales ($i>0$), the modules operate as *imputation models*, where each $\mathcal{F}^i_\theta$ refines missing fine-scale values by conditioning on both the current-scale input $\mathbf{X}^i$ and the aligned coarse predictions passed from previous stages. Formally,

$$\hat{\mathbf{Y}}^i = \mathcal{F}^i_\theta \Big( \text{concat}(\mathbf{X}^i, \hat{\mathbf{Y}}^{<i}_{\text{align}}) \Big), \qquad \hat{\mathbf{Y}}^i \in \mathbb{R}^{\frac{H}{d_i} \times C}. \tag{4}$$

This hierarchical design enforces a *scale-aligned information flow*: coarse predictions provide structural guidance, while finer-scale modules progressively enhance temporal resolution through imputation-based refinement.

Table 2: Long-term forecasting results averaged over four prediction lengths $\{96, 192, 336, 720\}$ with input length 336. Lower is better. Full results are listed in Appendix B.

| Dataset | SimpleTM | | + MSAR | | DUET | | + MSAR | | iTransformer | | +MSAR | | PatchTST | | + MSAR | | DLinear | | + MSAR | |
|---|---|---|---|---|---|---|---|---|---|---|---|---|---|---|---|---|---|---|---|---|
| | MSE | MAE | MSE | MAE | MSE | MAE | MSE | MAE | MSE | MAE | MSE | MAE | MSE | MAE | MSE | MAE | MSE | MAE | MSE | MAE |
| Electricity | **0.164** | **0.254** | 0.168 | 0.259 | 0.178 | 0.277 | **0.177** | **0.276** | 0.167 | 0.257 | **0.160** | **0.252** | 0.168 | 0.258 | **0.166** | **0.254** | 0.170 | 0.269 | 0.167 | 0.259 |
| ETTh1 | 0.424 | 0.437 | **0.410** | **0.424** | 0.403 | 0.415 | 0.404 | **0.415** | 0.475 | 0.459 | **0.449** | **0.455** | 0.428 | 0.439 | **0.413** | **0.427** | 0.420 | 0.429 | **0.419** | **0.426** |
| ETTh2 | 0.377 | 0.401 | **0.351** | **0.388** | 0.355 | 0.390 | **0.345** | **0.384** | 0.405 | 0.420 | **0.395** | **0.419** | 0.404 | 0.422 | **0.347** | **0.384** | 0.397 | 0.414 | **0.391** | **0.411** |
| ETTm1 | 0.354 | 0.375 | **0.345** | **0.371** | 0.352 | 0.370 | **0.351** | **0.369** | 0.388 | 0.394 | **0.359** | **0.384** | 0.371 | 0.379 | **0.345** | **0.368** | 0.354 | 0.370 | **0.353** | **0.369** |
| ETTm2 | 0.283 | 0.322 | **0.256** | **0.307** | 0.255 | 0.308 | **0.255** | **0.308** | 0.280 | 0.335 | **0.273** | **0.328** | 0.258 | 0.314 | **0.256** | **0.309** | 0.260 | 0.314 | **0.259** | **0.313** |
| Exchange | 0.438 | 0.432 | **0.352** | **0.401** | 0.394 | 0.422 | **0.372** | **0.408** | **0.403** | **0.444** | 0.686 | 0.538 | 0.465 | 0.462 | **0.364** | **0.408** | 0.524 | 0.455 | **0.371** | **0.406** |
| Traffic | **0.432** | 0.295 | 0.433 | **0.294** | 0.444 | 0.302 | 0.445 | **0.302** | 0.427 | 0.292 | **0.418** | **0.282** | 0.426 | 0.276 | **0.425** | **0.267** | 0.445 | 0.308 | 0.452 | 0.278 |
| Weather | 0.226 | 0.254 | **0.226** | **0.253** | 0.248 | 0.272 | **0.246** | **0.271** | 0.244 | 0.270 | **0.226** | **0.262** | 0.233 | 0.256 | **0.230** | **0.253** | 0.246 | 0.278 | **0.244** | **0.275** |
| Solar-Energy | 0.270 | 0.292 | **0.242** | **0.267** | 0.260 | 0.245 | **0.250** | **0.241** | 0.215 | 0.227 | **0.196** | **0.225** | 0.213 | 0.231 | **0.206** | **0.229** | 0.254 | 0.315 | 0.269 | **0.236** |
| Wind | 0.783 | 0.687 | **0.769** | **0.676** | 0.768 | 0.675 | **0.767** | **0.675** | 0.780 | 0.689 | **0.732** | **0.663** | 0.776 | 0.681 | **0.761** | **0.675** | 0.749 | 0.676 | 0.750 | **0.676** |

**Refinement in the Imputation Phase.** To further enhance cross-scale consistency, MSAR performs complete prediction over the target window $\mathbf{Y}^i$ during the imputation phase. Instead of restricting supervision to only the masked positions (mask = 0), we compute the loss across all target positions in the prediction window, including those that were previously filled (mask = 1). This approach enables iterative refinement, ensuring that coarse-scale predictions are smoothly integrated and corrected at finer resolutions, while maintaining temporal consistency across scales. This simple yet effective method adjusts previous predictions without introducing any additional parameters, helping to mitigate error accumulation across scales.

## 3.3 TRAINING STRATEGY

We train MSAR using the scale-wise MAE loss: $\mathcal{L} = \sum_{i=1}^{N} \text{MAE}(\hat{\mathbf{Y}}^i, \mathbf{Y}^i)$. However, directly optimizing this objective with fully autoregressive inputs leads to slow convergence and exposure bias—i.e., a discrepancy between training (teacher-forced) and inference-time inputs—commonly referred to as the teacher forcing problem (Williams & Zipser, 1989). To address this, we adopt a two-stage curriculum that gradually transitions from ground-truth conditioning to inference-aligned prediction.

**Teacher Forcing Training.** In the first stage, each scale-specific module $\mathcal{F}_\theta^i$ is trained independently using clean contextual signals. The coarsest forecaster ($i = 0$) is optimized directly on ground truth. For finer scales ($i > 0$), the model concatenates the ground-truth values of $\mathbf{Y}_{\text{align}}^{<i}$ from coarser levels with the current input. This teacher-forced approach stabilizes early learning and accelerates convergence by preventing error propagation across scales. Importantly, during this phase, the imputation models at finer scales use the ground-truth $Y$ values (rather than previous predictions), ensuring the model learns directly from the real target values without introducing refinement.

**Joint Training.** In the second stage, we switch to inference-aligned inputs: each imputation module receives predictions from coarser levels rather than ground truth, Crucially, during this phase we compute the loss across the *entire forecast horizon* at each scale (i.e., all $\frac{H}{d_i}$ steps), rather than restricting supervision to unobserved positions. This full-horizon supervision enables progressive refinement: predictions inherited from coarse scales can be corrected at finer resolutions, ensuring temporal coherence across scales. Thanks to its modular and scale-independent design, this curriculum allows MSAR to transition smoothly from robust pretraining under teacher forcing to deployment-consistent autoregressive inference.

## 4 EXPERIMENTS

We conduct extensive experiments to evaluate the effectiveness of the proposed Multi-Scale Autoregressive (MSAR) framework. Our empirical study is designed to address the following key questions: (1). Model-agnostic gains: As a plugin method, can MSAR consistently improve mainstream state-of-the-art (SOTA) forecasting models, and how does it compare against strong baselines? (cf. Sec. 4.1) (2). Efficiency study: What is the training and inference overhead introduced by MSAR compared to vanilla models? (cf. Sec. 4.2) (3). Mechanism analysis: Why does MSAR work? We perform ablation studies to dissect the contribution of its coarse-to-fine autoregressive design. (cf. Sec. 4.3) (4). Extended context: Can MSAR effectively benefit from longer look-back windows and better exploit historical dependencies? (cf. Sec. 4.4) (5). Sensitivity analysis: How do the number of autoregressive layers and the choice of interval sizes affect forecasting accuracy? (cf. Sec. 4.4)

Table 3: Multivariate forecasting results with prediction lengths $S \in \{96, 192, 336, 720\}$ for all and fixed lookback length $T = 336$. Results are averaged from all prediction lengths. *Avg* means further averaged by subsets. Full results are listed in Table 9.

| Models | MSAR (Ours) | | SimpleTM (2025) | | DUET (2025) | | xPatch (2025) | | PatchMLP (2025) | | iTransformer (2024a) | | TimeMixer (2024a) | | PatchTST (2023) | | LightTS (2023) | | DLinear (2023) | | FreTS (2023) | |
|---|---|---|---|---|---|---|---|---|---|---|---|---|---|---|---|---|---|---|---|---|---|---|
| Metric | MSE | MAE | MSE | MAE | MSE | MAE | MSE | MAE | MSE | MAE | MSE | MAE | MSE | MAE | MSE | MAE | MSE | MAE | MSE | MAE | MSE | MAE |
| ECL | **0.160** | **0.252** | 0.164 | 0.254 | 0.178 | 0.277 | 0.170 | 0.264 | 0.200 | 0.301 | 0.167 | 0.257 | 0.173 | 0.267 | 0.168 | 0.258 | 0.207 | 0.316 | 0.170 | 0.269 | 0.171 | 0.270 |
| ETT (Avg) | **0.335** | **0.367** | 0.359 | 0.383 | 0.341 | 0.370 | 0.352 | 0.387 | 0.371 | 0.399 | 0.387 | 0.402 | 0.352 | 0.385 | 0.365 | 0.393 | 0.460 | 0.454 | 0.357 | 0.381 | 0.402 | 0.419 |
| Exchange | **0.327** | **0.385** | 0.438 | 0.432 | 0.394 | 0.422 | 0.940 | 0.707 | 0.370 | 0.413 | 0.403 | 0.444 | 0.354 | 0.414 | 0.465 | 0.462 | 0.518 | 0.429 | 0.524 | 0.455 | 0.613 | 0.539 |
| Traffic | **0.413** | **0.267** | 0.432 | 0.295 | 0.444 | 0.302 | 0.429 | 0.299 | 0.524 | 0.382 | 0.427 | 0.292 | 0.444 | 0.316 | 0.426 | 0.276 | 0.507 | 0.369 | 0.447 | 0.301 | 0.461 | 0.313 |
| Weather | **0.223** | **0.252** | 0.226 | 0.254 | 0.248 | 0.272 | 0.224 | 0.262 | 0.230 | 0.267 | 0.244 | 0.270 | 0.226 | 0.266 | 0.233 | 0.256 | 0.230 | 0.281 | 0.246 | 0.278 | 0.229 | 0.276 |
| Solar-Energy | **0.195** | **0.223** | 0.270 | 0.292 | 0.260 | 0.245 | 0.200 | 0.253 | 0.250 | 0.292 | 0.215 | 0.227 | 0.218 | 0.276 | 0.213 | 0.231 | 0.217 | 0.288 | 0.255 | 0.315 | 0.212 | 0.267 |
| Wind | 0.732 | **0.663** | 0.783 | 0.687 | 0.768 | 0.675 | 0.762 | 0.682 | 0.772 | 0.689 | 0.780 | 0.689 | 0.762 | 0.684 | 0.776 | 0.681 | **0.726** | 0.670 | 0.749 | 0.676 | 0.742 | 0.679 |

**Dataset.** For the long-term forecasting experiments, we utilize a diverse set of datasets to evaluate the robustness and generalizability of our models across different domains. These datasets include ECL, ETT (4 subsets), Exchange, Traffic, Weather, Solar-Energy, and Wind [2]. Additionally, for the short-term forecasting experiments, the PEMS (4 subsets) (Chen et al., 2001) dataset is employed, specifically designed for short-term traffic prediction tasks. Following TSLib (Wang et al., 2024b), we adopt the same dataset splits, data preprocessing steps and ensure no last batch is dropped during training. For long-term forecasting, we use prediction lengths of {96, 192, 336, 720}, while for short-term forecasting, we use a prediction length of 12.

**Baselines.** To evaluate the performance of our proposed MSAR framework, we compare it against several state-of-the-art (SOTA) methods, which include SimpleTM (Chen et al., 2025), DUET (Qiu et al., 2025), iTransformer (Liu et al., 2024a), PatchTST (Nie et al., 2023), and DLinear (Qiu et al., 2024a). Notably, DUET has achieved the best performance on the TFB benchmark (Qiu et al., 2024b), underscoring its effectiveness in handling time-series forecasting tasks. In addition to evaluating MSAR as a plug-in on the above models, we also validate its performance gain by directly comparing it with additional methods including xPatch (Stitsyuk & Choi, 2025), PatchMLP (Tang & Zhang, 2025), TimeMixer (Wang et al., 2024a), LightTS (Campos et al., 2023), and FreTS (Yi et al., 2023).

**Unified experiment settings.** To fairly evaluate the effectiveness of MSAR, we designed three experimental setups: (1). **Baseline vs. Baseline + MSAR**: In this experiment, we compare the performance of the baseline models with and without MSAR, ensuring that *all hyperparameters remain consistent across both setups*. (2). **One-to-Many Comparison**: We select the top-ranked model from the TFB (Qiu et al., 2024b) benchmark and incorporate MSAR into it. This setup is compared against other state-of-the-art (SOTA) models, while ensuring that all core parameters are consistent across models. (3). **Full Hyperparameter Search**: For this experiment, we perform a comprehensive search over the full hyperparameter space to ensure that all models are evaluated with the same range of parameter values, allowing for a fair comparison. Detailed experimental settings can be found in the Appendix A.3.

## 4.1 FORECASTING RESULTS

Table 2 reports average results across four horizons 96, 192, 336, 720 with all methods extracting features from a look-back window of length 336. Across all eight datasets and five backbones, equipping MSAR consistently improves performance. For instance, DLinear's MSE on Traffic drops from 0.432 to 0.419, and PatchTST's MAE on Weather decreases from 0.256 to 0.250. These consistent gains across both lightweight linear models and Transformer-style architectures (PatchTST, iTransformer, DUET, SimpleTM) highlight the generality of MSAR as a plug-in framework. Full per-horizon results are given in Appendix B.

To ensure a fair and comprehensive evaluation, we adopt two complementary comparison settings. In Table 3, we directly integrate MSAR into the top-ranked backbone identified by the TFB bench-

---

[2]https://www.kaggle.com/datasets/mubashirrahim/wind-power-generation-data-forecasting

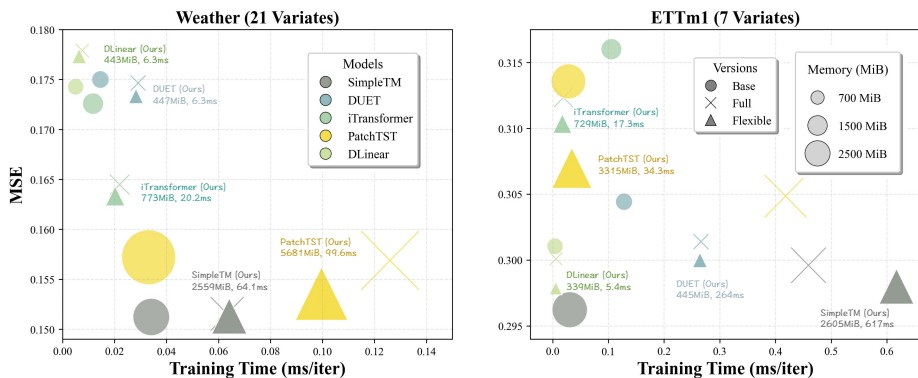

Figure 3: Efficiency–accuracy trade-off on **Weather** (21 variates) and **ETTm1** (7 variates).

mark and compare it against other state-of-the-art methods under identical experimental configurations. We further report results under a full hyperparameter search (detailed in Appendix 10). This setting allows each model to be evaluated under its best-performing configuration within a shared search space, thereby reflecting the optimal modeling capability of different approaches.

**Stratified Performance Analysis.** To further contextualize the magnitude and consistency of improvements, we conduct a stratified analysis using dataset meta-features reported in TFB (Qiu et al., 2024b) benchmark, including seasonality strength, trend/non-stationarity, noise ratio, input dimensionality, and effective horizon difficulty. Three observations emerge.

❶ **Larger gains on datasets with strong periodicity and stable seasonal structure.** ETTh1/ETTh2 and ETTm1/ETTm2 are categorized as highly periodic and low-to-moderate noise. MSAR achieves the largest improvements here across nearly all backbones. Coarse scales reliably capture long-range seasonal cycles, providing stable guidance for finer-scale refinement.

❷ **MSAR is particularly effective for long horizons where long-range structure dominates.** The strongest improvements appear at horizons 336 and 720 on the ETT series and Wind datasets. This is consistent with the coarse-to-fine design: coarse-scale forecasting models slow-varying patterns, while finer scales reconstruct higher-frequency details.

❸ **Moderate but consistent gains on high-noise or weakly periodic datasets.** Traffic—characterized by high stochasticity and weak periodicity—shows smaller but still stable improvements. In these settings, coarse-scale signals carry limited structure, and irregular fine-scale fluctuations dominate; MSAR nevertheless improves long-horizon accuracy by stabilizing coarse-level guidance.

### 4.2 EFFICIENCY STUDY

Beyond forecasting accuracy, we further evaluate the *efficiency* of MSAR. Figure 6 plots training time (x-axis), error (y-axis), and GPU memory (bubble size) for five representative baselines and their MSAR-augmented counterparts on Weather and ETTm1 ($H$=336, $T$=192). We report two variants of MSAR: *full* and *flexible*.

The *full* variant applies a unified look-back length of 336 for both forecasting and imputation. In this case, the forecasting module consumes a compressed 56-length input (via 6× downsampling), while the imputation module operates on the full 336 tokens. The *flexible* variant instead shortens the imputation look-back to 192, exploiting MSAR's plug-in, model-agnostic design. This simple change leads to a substantial reduction in training cost while often *improving* accuracy, as shown in Figure 6.

MSAR introduces additional training-time overhead, primarily because of the two-stage training strategy designed to mitigate the train–test discrepancy caused by teacher forcing. However, this cost is offset by two practical advantages: (i) MSAR yields more stable convergence and improves inference-time behavior; (ii) its decoupled per-scale forecasters significantly reduce memory usage

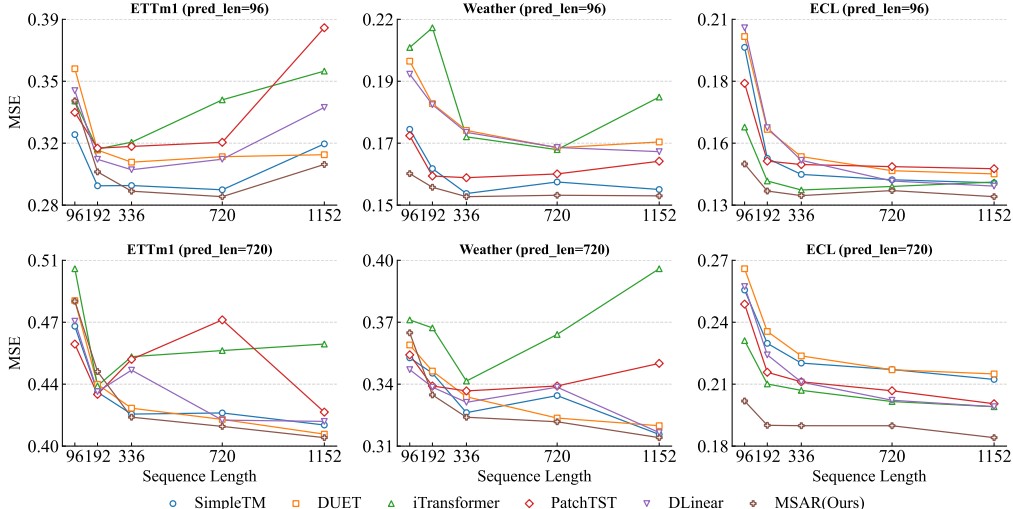

Figure 4: Impact of look-back window length on forecasting performance (measured by MSE). We report results on three representative datasets (ETTm1, Weather, and Electricity) under two horizons ($H = 96$ and $H = 720$). Each curve shows the error as the input sequence length increases from 96 to 1152.

Table 4: Ablation study on MSAR components over nine datasets. ① = Multiscale, ② = Alignment, ③ = Scale-wise AR. Each row shows the combination of enabled components. Results with * denote training without the joint training phase, while results with † denote removing the full pred-len imputation loss.

| Setting | ECL | | ETTm1 | | ETTm2 | | ETTh1 | | ETTh2 | | Traffic | | Weather | | Solar-Energy | | Wind | |
|---|---|---|---|---|---|---|---|---|---|---|---|---|---|---|---|---|---|---|
| | MSE | MAE | MSE | MAE | MSE | MAE | MSE | MAE | MSE | MAE | MSE | MAE | MSE | MAE | MSE | MAE | MSE | MAE |
| ① | 0.198 | 0.290 | 0.405 | 0.410 | 0.287 | 0.342 | 0.452 | 0.460 | 0.432 | 0.446 | 0.455 | 0.300 | 0.246 | 0.272 | 0.236 | 0.252 | 0.790 | 0.695 |
| ② | 0.168 | 0.258 | 0.371 | 0.379 | 0.258 | 0.314 | 0.428 | 0.439 | 0.404 | 0.422 | 0.426 | 0.276 | **0.223** | 0.256 | 0.213 | 0.231 | 0.761 | 0.675 |
| ③ | 0.190 | 0.282 | 0.395 | 0.402 | 0.280 | 0.335 | 0.445 | 0.452 | 0.426 | 0.438 | 0.448 | 0.293 | 0.242 | 0.266 | 0.230 | 0.245 | 0.782 | 0.690 |
| ①+② | 0.183 | 0.276 | 0.385 | 0.392 | 0.273 | 0.327 | 0.438 | 0.446 | 0.418 | 0.430 | 0.438 | 0.286 | 0.238 | 0.260 | 0.224 | 0.238 | 0.774 | 0.684 |
| ①+③ | 0.188 | 0.280 | 0.392 | 0.399 | 0.277 | 0.331 | 0.442 | 0.449 | 0.422 | 0.434 | 0.442 | 0.289 | 0.240 | 0.263 | 0.227 | 0.241 | 0.778 | 0.687 |
| ①+②+③ | **0.166** | **0.254** | **0.345** | **0.368** | **0.256** | **0.309** | **0.413** | **0.427** | **0.347** | **0.384** | **0.425** | **0.267** | 0.230 | **0.253** | **0.206** | **0.229** | **0.761** | **0.675** |
| ①+②+③* | 0.172 | 0.262 | 0.375 | 0.383 | 0.262 | 0.318 | 0.431 | 0.443 | 0.408 | 0.426 | 0.430 | 0.280 | 0.233 | 0.260 | 0.216 | 0.235 | 0.765 | 0.679 |
| ①+②+③† | 0.176 | 0.266 | 0.379 | 0.387 | 0.266 | 0.322 | 0.435 | 0.447 | 0.412 | 0.430 | 0.434 | 0.284 | 0.237 | 0.264 | 0.220 | 0.239 | 0.769 | 0.683 |

compared to conventional autoregressive methods that must unroll full-sequence dependencies. As a result, MSAR can be seamlessly integrated into diverse backbone architectures while offering a favorable efficiency–accuracy trade-off and strong overall performance.

We further evaluate inference-time efficiency in Appendix C. MSAR reduces inference latency for PatchTST (Nie et al., 2023), DLinear (Zeng et al., 2023), and TimesNet (Wu et al., 2022)—often by nearly a factor of two—because scale-wise forecasting shortens the effective sequence length processed at the fine scale. For iTransformer (Liu et al., 2024a), MSAR introduces only a slight overhead, which is expected given its variate-wise embedding over the full sequence, where reducing token count does not yield proportional savings.

## 4.3 ABLATION STUDY

In this section, we ask: *where does the performance gain of MSAR come from?* We ablate three design components: ① Multiscale, ② Alignment modeling in encoder, and ③ Scale-wise Autoregression (AR), using PatchTST (Nie et al., 2023) as the backbone. Alignment enforces cross-scale consistency between the input $X$ and output $Y$.

Table 4 shows that none of the components alone is sufficient. ① provides additional context but suffers from scale mixing; ② behaves similarly to the single-scale baseline; and ③ resembles coarse-to-fine AR methods (e.g., C2FAR (Bergsma et al., 2023b)), but lacks input–output alignment. Pairwise

combinations offer partial gains, but the behavior reflects the intrinsic dependencies among components. ①+② improves upon ① because alignment removes the scale-mix-up effect in the encoder, but it remains weaker than ②: multiscale modeling produces duplicated predictions on overlapping time indices, forcing the decoder to fuse them and thereby reintroducing mild misalignment. ①+③ also improves over single components, but is weaker than ② because multiscale inputs and cross-scale supervision break the $X$–$Y$ temporal granularity, making optimization more difficult.

When all components are combined (①+②+③), MSAR eliminates encoder-side scale mixing, preserves strict input–output alignment, and enables clean coarse-to-fine refinement without any cross-scale fusion. This full design consistently yields the best performance across all datasets, demonstrating that MSAR's improvements arise from the **synergy** among multiscale representation, scale-aligned modeling, and scale-wise autoregressive refinement.

### 4.4 MODEL ANALYSIS

**Sensitivity Analysis.** We further investigate the sensitivity of MSAR with respect to the choice of interval list and the number of layers on the ETTh1 dataset. As shown in Figure 5, the top panel reports the average performance of five backbones under different interval lists. The results remain stable across varying downsampling factors, indicating that MSAR is largely insensitive to the specific interval choices and thus requires minimal tuning. The bottom panel illustrates the impact of the number of layers. While adding one or two imputation layers consistently improves both MSE and MAE over the baseline, further increasing the number of layers yields diminishing returns and in some cases even slight degradation. These observations confirm that MSAR achieves robust performance across settings, with a favorable trade-off between accuracy and complexity at shallow depths.

**Extended context.** To further examine the ability of MSAR to utilize long historical contexts, we study the impact of increasing the look-back window length $T$ from 96 to 1152 under two horizons ($H$=96 and $H$=720). Figure 4 reports results on three representative datasets (ETTm1, Weather, and Electricity). This analysis validates that MSAR effectively leverages extended historical sequences. Unlike existing models, which risk overfitting or noise amplification with longer inputs, MSAR transforms additional context into meaningful coarse-to-fine supervision, achieving consistent improvements in both short- and long-horizon forecasting.

**Limitations and Future Work.** Despite the strong empirical performance and broad applicability of the proposed MSAR framework, several limitations remain. First, while MSAR decouples temporal patterns across scales and achieves consistent gains across diverse backbones, its hierarchical information flow may still propagate accumulation errors from coarse to fine resolutions. Second, MSAR introduces additional computational stages compared to single-scale baselines. Although we provide a flexible variant that reduces overhead with negligible accuracy loss, scaling to ultra-long sequences or real-time applications may require more efficient scheduling or adaptive scale selection. For future work, a potential improvement is to partially retain inputs from the previous stage and selectively mask them with structured noise (Chen et al., 2024a). By treating noise as a form of adaptive masking, the model could learn to denoise and refine inherited predictions, improving robustness against propagated errors while maintaining consistency across scales.

## 5 CONCLUSIONS

In this study, we propose the Modular Scale-wise Autoregressive Framework (MSAR) to overcome the limitations of existing time series forecasting methods that suffer from scale misalignment and limited long-context utilization. Unlike prior approaches that apply multiscale modeling only to the input sequence or rely on single-scale decoding, MSAR performs fully scale-aligned modeling across both inputs and outputs, enabling each scale to be modeled independently. By introducing a scale-wise autoregressive strategy, MSAR progressively refines predictions from coarse to fine resolutions through an efficient, modular information flow. Our work highlights the importance of aligning temporal granularity in both modeling and generation, paving the way for scalable and robust multiscale forecasting.

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

## A  IMPLEMENTATION DETAILS

### A.1  DATASET DESCRIPTIONS

We conduct experiments on **seventeen** widely used real-world multivariate time series datasets, covering both **long-term** and **short-term** forecasting tasks. The datasets span diverse domains including electricity, weather, traffic, and renewable energy, ensuring the robustness and generality of our approach. Table 5 summarizes the statistics of all datasets. Specifically:

- **Electricity Transformer Temperature (ETT)** (Zhou et al., 2021): Four subsets are provided. ETTh1/ETTh2 are hourly, while ETTm1/ETTm2 are recorded every 15 minutes. All series are collected from two transformers.

- **Weather** (Wu et al., 2021): 21 meteorological indicators collected every 10 minutes in Germany during 2020 from the Max Planck Biogeochemistry Institute.

- **Traffic** (Wu et al., 2021): Hourly road occupancy rates from 862 sensors on San Francisco Bay Area freeways (2015–2016).

- **Electricity** (Wu et al., 2021): Hourly consumption of 321 clients from 2012–2014.

- **Exchange-rate** (Wu et al., 2021): Daily exchange rates of 8 countries from 1990–2016.

- **Solar-energy** Lai et al. (2018): 10-minute solar power production from 137 photovoltaic plants in 2006.

- **Wind**[3]: Hourly wind speed and power generation from four real-world wind farms.

- **PEMS**: High-frequency traffic measurements with 5-minute resolution. We use four subsets (PEMS03, PEMS04, PEMS07, and PEMS08), each corresponding to different spatial sensor networks.

Table 5: Dataset statistics. "Dim" denotes the number of variates; "Dataset Size" shows the number of time points in (Train/Validation/Test) splits; "Frequency" is the sampling interval.

| Dataset | Dim | Dataset Size (Train/Val/Test) | Frequency |
|---|---|---|---|
| ETTh1, ETTh2 | 7 | (8545, 2881, 2881) | Hourly |
| ETTm1, ETTm2 | 7 | (34465, 11521, 11521) | 15 min |
| Weather | 21 | (36792, 5271, 10540) | 10 min |
| Traffic | 862 | (12185, 1757, 3509) | Hourly |
| Electricity | 321 | (18317, 2633, 5261) | Hourly |
| Exchange-rate | 8 | (5120, 665, 1422) | Daily |
| Solar-energy | 137 | (36792, 5271, 10540) | 10 min |
| Wind | 9 | (30468, 4286, 8665) | Hourly |
| PEMS03 | 358 | (15617, 5135, 5135) | 5 min |
| PEMS04 | 307 | (10172, 3375, 3375) | 5 min |
| PEMS07 | 883 | (16911, 5622, 5622) | 5 min |
| PEMS08 | 170 | (10690, 3548, 265) | 5 min |

### A.2  EVALUATION METRICS

We follow standard practices in time series forecasting and adopt different metrics for short-term and long-term datasets.

**Short-term forecasting.** For short-term benchmarks such as PEMS, we evaluate models using three complementary error measures:

- Mean Absolute Error (MAE): $\frac{1}{T} \sum_{t=1}^{T} |\hat{y}_t - y_t|$

- Mean Absolute Percentage Error (MAPE): $\frac{1}{T} \sum_{t=1}^{T} \frac{|\hat{y}_t - y_t|}{|y_t|}$

---

[3]https://www.kaggle.com/datasets/mubashirrahim/wind-power-generation-data-forecasting

- Root Mean Squared Error (RMSE): $\sqrt{\frac{1}{T}\sum_{t=1}^{T}(\hat{y}_t - y_t)^2}$

MAE reflects absolute deviation, MAPE measures relative deviation with respect to ground truth magnitude, and RMSE penalizes large errors more heavily. Together, these metrics provide a comprehensive evaluation of accuracy and robustness.

**Long-term forecasting.** For long-term benchmarks such as ETT, Weather, and Exchange, we report Mean Squared Error (MSE) and Mean Absolute Error (MAE):

- Mean Squared Error (MSE): $\frac{1}{T}\sum_{t=1}^{T}(\hat{y}_t - y_t)^2$

- Mean Absolute Error (MAE): $\frac{1}{T}\sum_{t=1}^{T}|\hat{y}_t - y_t|$

These metrics are widely adopted in prior work (Nie et al., 2023), and capture both variance-sensitive and absolute error perspectives. MSE emphasizes stability for long horizons by penalizing large deviations, while MAE provides scale-consistent interpretability. All metrics are averaged across all variates and prediction horizons. Lower values indicate better forecasting performance.

### A.3 EXPERIMENT DETAILS

To fairly evaluate the effectiveness of MSAR, we design three experimental setups: (1) **Baseline vs. Baseline+MSAR**: we compare the performance of each baseline model with and without MSAR, while keeping *all hyperparameters consistent* across both settings. (2) **One-to-Many Comparison**: we select the top-ranked model from the TFB (Qiu et al., 2024b) benchmark and incorporate MSAR into it, then compare against other SOTA methods with matched core parameters. (3) **Full Hyperparameter Search**: we perform a comprehensive grid search over the full hyperparameter space so that all models are tuned under the same parameter search range.

**Unified Settings for (1) and (2).** For the first two setups, we adopt a unified input length (`seq_len` = 336) across all datasets, with the detailed hyperparameters summarized in Table 7. For MSAR, we fix the interval list to `interval_list` = $[6, 1]$, which corresponds to `seqlen_list` = $[56, 336]$ for coarse- and fine-scale autoregression. This configuration guarantees that the baseline models and their MSAR-augmented counterparts are evaluated under identical input conditions, thereby ensuring a fair comparison.

**Setup for (3).** For the full hyperparameter search, all methods are tuned under an identical search space to ensure fairness. Specifically, the candidate ranges are:

- `seq_len` $\in \{96, 192, 320, 512\}$,

- `d_model` $\in \{16, 128, 512\}$,

- `e_layers` $\in \{1, 3, 5\}$,

- `epoch` $\in \{50\}$,

- learning rate $\in \{1\times 10^{-5}, 1\times 10^{-4}, 1\times 10^{-3}, 1\times 10^{-2}, 0.05\}$.

Other hyperparameters are fixed across all models, including `batch_size`= 32, `n_heads`= 8, and `d_layers`= 1.

This unified search space covers model width, depth, horizon length, and optimization settings, and is applied consistently across all baselines. Therefore, the reported results reflect the intrinsic modeling capability of each method rather than differences in hyperparameter tuning. Table 6 reports the full hyperparameter configurations obtained from our unified search space for all datasets and prediction lengths.

All experiments are implemented in Pytorch (Paszke et al., 2017), and conducted on two machines, each equipped with four NVIDIA 4090 GPUs.

Table 6: Best hyperparameter configurations identified in MSAR.

| Dataset | pred_len | $d_{model}$ | $d_{ff}$ | Layers | LR* | s_list |
|---|---|---|---|---|---|---|
| ETTh1 | 96 | 512 | 2048 | 5 | $1 \times 10^{-3}$ | [32, 512] |
| | 192 | 512 | 2048 | 5 | $1 \times 10^{-3}$ | [16, 512] |
| | 336 | 128 | 512 | 1 | $1 \times 10^{-3}$ | [32, 320] |
| | 720 | 128 | 512 | 5 | $1 \times 10^{-3}$ | [85, 192] |
| ETTh2 | 96 | 512 | 2048 | 1 | $1 \times 10^{-3}$ | [16, 512] |
| | 192 | 128 | 512 | 5 | $1 \times 10^{-3}$ | [16, 512] |
| | 336 | 512 | 2048 | 5 | $1 \times 10^{-3}$ | [32, 192] |
| | 720 | 512 | 2048 | 1 | $1 \times 10^{-3}$ | [16, 320] |
| ETTm1 | 96 | 128 | 512 | 5 | $1 \times 10^{-3}$ | [85, 512] |
| | 192 | 512 | 2048 | 5 | $1 \times 10^{-3}$ | [32, 320] |
| | 336 | 16 | 64 | 1 | $5 \times 10^{-2}$ | [32, 192] |
| | 720 | 128 | 512 | 5 | $1 \times 10^{-3}$ | [32, 512] |
| ETTm2 | 96 | 128 | 512 | 3 | $1 \times 10^{-3}$ | [32, 512] |
| | 192 | 16 | 64 | 1 | $5 \times 10^{-2}$ | [85, 512] |
| | 336 | 16 | 64 | 1 | $5 \times 10^{-2}$ | [85, 320] |
| | 720 | 128 | 512 | 3 | $1 \times 10^{-3}$ | [32, 512] |
| Weather | 96 | 512 | 2048 | 5 | $1 \times 10^{-3}$ | [16, 320] |
| | 192 | 128 | 512 | 3 | $1 \times 10^{-3}$ | [16, 512] |
| | 336 | 512 | 2048 | 5 | $1 \times 10^{-3}$ | [16, 192] |
| | 720 | 16 | 64 | 1 | $5 \times 10^{-2}$ | [32, 512] |

∗ LR means the initial learning rate.

Table 7: Experiment configuration of MSAR in Table 8. All experiments use the Adam optimizer with a learning rate of $10^{-3}$ and early stopping (patience = 3).

| Dataset / Configurations | Model Hyper-parameter | | | | Training Process | | | |
|---|---|---|---|---|---|---|---|---|
| | $d_{model}$ | $d_{ff}$ | Layers | Heads | Batch Size | Epochs | LR* | Patience |
| ECL | 128 | 256 | 1 | 2 | 16 | 10 | $10^{-3}$ | 3 |
| Traffic | 128 | 256 | 1 | 2 | 16 | 10 | $10^{-3}$ | 3 |
| Solar | 128 | 256 | 1 | 2 | 16 | 10 | $10^{-3}$ | 3 |
| ETTh1 | 512 | 2048 | 2 | 8 | 32 | 10 | $10^{-3}$ | 3 |
| ETTh2 | 512 | 2048 | 2 | 8 | 32 | 10 | $10^{-3}$ | 3 |
| ETTm1 | 512 | 2048 | 2 | 8 | 32 | 10 | $10^{-3}$ | 3 |
| ETTm2 | 512 | 2048 | 2 | 8 | 32 | 10 | $10^{-3}$ | 3 |
| Weather | 512 | 2048 | 2 | 8 | 32 | 10 | $10^{-3}$ | 3 |
| Exchange | 512 | 2048 | 2 | 8 | 32 | 10 | $10^{-3}$ | 3 |
| Wind | 512 | 2048 | 2 | 8 | 32 | 10 | $10^{-3}$ | 3 |

∗ LR means the initial learning rate.

## B  MORE RESULTS

**Full Results in Table 2.** Table 8 reports the *full* long-term forecasting results across all predic-tion lengths $\{96, 192, 336, 720\}$ on the benchmark datasets. Here we provide the complete results to ensure full transparency and allow a more fine-grained comparison across different forecasting lengths.

**Full Results in Table 3.** Table 3 reports the *full* long-term forecasting results across all prediction lengths $\{96, 192, 336, 720\}$ on the benchmark datasets. Our results correspond to MSAR+DUET.

Table 8: Full Long-term forecasting results on 10 datasets.

| Models | Metric | SimpleTM | | | | DUET | | | | iTransformer | | | | PatchTST | | | | DLinear | | | |
|---|---|---|---|---|---|---|---|---|---|---|---|---|---|---|---|---|---|---|---|---|---|
| | | Base | | +Ours | | Base | | +Ours | | Base | | +Ours | | Base | | +Ours | | Base | | +Ours | |
| | | MSE | MAE | MSE | MAE | MSE | MAE | MSE | MAE | MSE | MAE | MSE | MAE | MSE | MAE | MSE | MAE | MSE | MAE | MSE | MAE |
| Electricity | 96 | **0.134** | **0.226** | 0.137 | 0.231 | 0.149 | 0.251 | **0.147** | **0.250** | 0.135 | 0.229 | **0.132** | **0.226** | 0.138 | 0.231 | **0.136** | **0.227** | 0.143 | 0.242 | **0.141** | **0.234** |
| | 192 | 0.154 | 0.245 | **0.152** | **0.244** | 0.163 | 0.264 | **0.162** | **0.263** | 0.153 | 0.245 | **0.149** | **0.241** | 0.154 | 0.246 | **0.152** | **0.241** | 0.157 | 0.255 | **0.155** | **0.246** |
| | 336 | 0.175 | 0.268 | **0.167** | **0.258** | 0.181 | 0.281 | **0.179** | **0.279** | 0.169 | 0.261 | **0.164** | **0.257** | 0.171 | 0.261 | **0.168** | **0.257** | 0.172 | 0.272 | **0.170** | **0.262** |
| | 720 | **0.191** | **0.278** | 0.214 | 0.303 | 0.220 | 0.311 | **0.219** | **0.310** | 0.209 | 0.294 | **0.196** | **0.285** | 0.210 | 0.294 | **0.207** | **0.290** | 0.207 | 0.305 | **0.204** | **0.293** |
| | Avg | **0.164** | **0.254** | 0.168 | 0.259 | 0.178 | 0.277 | **0.177** | **0.276** | 0.167 | 0.257 | **0.160** | **0.252** | 0.168 | 0.258 | **0.166** | **0.254** | 0.170 | 0.269 | **0.167** | **0.259** |
| ETTm1 | 96 | 0.291 | 0.337 | **0.282** | **0.330** | **0.292** | 0.335 | 0.293 | **0.335** | 0.318 | 0.355 | **0.300** | **0.344** | 0.317 | 0.343 | **0.280** | **0.327** | 0.293 | 0.333 | **0.293** | **0.333** |
| | 192 | 0.330 | 0.363 | **0.324** | **0.357** | 0.332 | 0.359 | **0.330** | **0.356** | 0.370 | 0.383 | **0.337** | **0.370** | 0.335 | 0.361 | **0.324** | **0.356** | 0.332 | 0.355 | **0.332** | **0.355** |
| | 336 | 0.369 | 0.384 | **0.357** | **0.381** | 0.365 | 0.378 | **0.363** | **0.376** | 0.381 | 0.384 | **0.372** | **0.393** | 0.381 | 0.384 | **0.361** | **0.379** | 0.366 | 0.376 | 0.366 | 0.377 |
| | 720 | 0.425 | 0.416 | **0.416** | **0.415** | 0.419 | 0.409 | 0.419 | 0.409 | 0.462 | 0.434 | **0.428** | **0.427** | 0.451 | 0.427 | **0.414** | **0.412** | 0.426 | 0.417 | **0.422** | **0.411** |
| | Avg | 0.354 | 0.375 | **0.345** | **0.371** | 0.352 | 0.370 | **0.351** | **0.369** | 0.388 | 0.394 | **0.359** | **0.384** | 0.371 | 0.379 | **0.345** | **0.368** | 0.354 | 0.370 | **0.353** | **0.369** |
| ETTm2 | 96 | 0.177 | 0.255 | **0.165** | **0.246** | 0.163 | 0.248 | **0.163** | **0.247** | 0.179 | 0.266 | **0.172** | **0.255** | 0.173 | 0.258 | **0.164** | **0.246** | 0.165 | 0.248 | **0.165** | **0.247** |
| | 192 | 0.251 | 0.302 | **0.221** | **0.287** | 0.219 | 0.285 | **0.219** | **0.285** | 0.254 | 0.323 | **0.228** | **0.298** | 0.219 | 0.288 | 0.224 | 0.288 | 0.222 | 0.288 | **0.222** | **0.288** |
| | 336 | 0.305 | 0.338 | **0.272** | **0.319** | 0.273 | 0.322 | **0.272** | 0.322 | 0.300 | 0.350 | **0.296** | 0.351 | 0.279 | 0.327 | **0.279** | **0.324** | 0.276 | 0.327 | **0.276** | **0.326** |
| | 720 | 0.399 | 0.394 | **0.364** | **0.378** | 0.365 | 0.379 | **0.364** | **0.379** | 0.389 | 0.399 | 0.397 | 0.409 | 0.361 | 0.381 | **0.355** | **0.379** | 0.375 | 0.393 | **0.371** | **0.391** |
| | Avg | 0.283 | 0.322 | **0.256** | **0.307** | 0.255 | 0.308 | **0.255** | **0.308** | 0.280 | 0.335 | **0.273** | **0.328** | 0.258 | 0.314 | **0.256** | **0.309** | 0.260 | 0.314 | **0.259** | **0.313** |
| ETTh1 | 96 | 0.379 | 0.401 | **0.361** | **0.389** | 0.359 | 0.383 | **0.358** | **0.381** | 0.409 | 0.406 | **0.387** | **0.406** | 0.383 | 0.403 | **0.359** | **0.386** | 0.365 | 0.383 | **0.365** | **0.383** |
| | 192 | 0.423 | 0.430 | **0.410** | **0.418** | 0.398 | 0.406 | **0.398** | **0.406** | 0.507 | 0.462 | **0.429** | **0.436** | 0.409 | 0.424 | **0.397** | **0.412** | 0.417 | 0.426 | **0.405** | **0.409** |
| | 336 | 0.440 | 0.443 | **0.428** | **0.431** | 0.426 | 0.422 | 0.428 | 0.423 | 0.459 | **0.451** | 0.457 | 0.457 | 0.440 | 0.446 | **0.422** | **0.433** | 0.443 | 0.436 | **0.440** | **0.431** |
| | 720 | 0.455 | 0.475 | **0.440** | **0.460** | **0.430** | **0.450** | 0.433 | 0.452 | 0.526 | **0.506** | **0.523** | 0.519 | 0.481 | 0.482 | **0.475** | **0.479** | 0.456 | 0.473 | 0.467 | 0.481 |
| | Avg | 0.424 | 0.437 | **0.410** | **0.424** | 0.403 | 0.415 | 0.404 | 0.415 | 0.475 | 0.459 | **0.449** | **0.455** | 0.424 | 0.439 | **0.413** | **0.427** | 0.420 | 0.429 | **0.419** | **0.426** |
| ETTh2 | 96 | 0.295 | 0.342 | **0.277** | **0.334** | 0.273 | 0.331 | **0.272** | **0.330** | 0.343 | 0.373 | **0.292** | **0.348** | 0.292 | 0.346 | **0.288** | **0.338** | 0.278 | 0.333 | **0.278** | **0.333** |
| | 192 | 0.361 | 0.388 | **0.347** | **0.379** | 0.347 | 0.382 | **0.338** | **0.375** | 0.410 | 0.413 | **0.370** | **0.399** | 0.354 | 0.385 | **0.350** | **0.378** | 0.350 | 0.382 | 0.352 | 0.385 |
| | 336 | 0.426 | 0.428 | **0.372** | **0.405** | 0.394 | 0.415 | **0.371** | **0.402** | 0.419 | 0.434 | 0.421 | 0.431 | 0.383 | 0.407 | **0.365** | **0.400** | 0.403 | 0.424 | **0.389** | **0.413** |
| | 720 | 0.427 | 0.446 | **0.408** | **0.432** | 0.405 | 0.433 | **0.400** | **0.430** | 0.448 | 0.461 | 0.497 | 0.498 | 0.587 | 0.549 | **0.387** | **0.420** | 0.556 | 0.518 | **0.515** | **0.515** |
| | Avg | 0.377 | 0.401 | **0.351** | **0.388** | 0.355 | 0.390 | **0.345** | **0.384** | 0.405 | 0.420 | **0.395** | **0.419** | 0.404 | 0.422 | **0.347** | **0.384** | 0.397 | 0.414 | **0.391** | **0.411** |
| Exchange | 96 | 0.094 | 0.216 | **0.090** | **0.212** | 0.092 | 0.213 | **0.088** | **0.206** | 0.109 | 0.239 | **0.107** | **0.237** | 0.098 | 0.221 | **0.090** | **0.210** | **0.082** | **0.201** | 0.095 | 0.214 |
| | 192 | 0.190 | 0.313 | **0.183** | **0.308** | 0.189 | 0.307 | **0.184** | **0.302** | **0.266** | **0.377** | 0.269 | 0.367 | 0.210 | 0.329 | **0.195** | **0.317** | 0.174 | 0.299 | **0.154** | **0.283** |
| | 336 | **0.331** | **0.417** | 0.342 | 0.424 | 0.392 | 0.456 | **0.354** | **0.428** | **0.396** | **0.467** | 0.579 | 0.566 | 0.468 | 0.506 | **0.379** | **0.443** | 0.347 | 0.427 | **0.276** | **0.392** |
| | 720 | 1.138 | 0.784 | **0.792** | **0.659** | 0.901 | 0.714 | **0.863** | **0.697** | 0.841 | 0.691 | **0.791** | **0.678** | 1.082 | 0.793 | **0.791** | **0.661** | 1.493 | 0.893 | **0.961** | **0.737** |
| | Avg | 0.438 | 0.432 | **0.352** | **0.401** | 0.394 | 0.422 | **0.372** | **0.408** | **0.403** | **0.444** | 0.686 | 0.538 | 0.465 | 0.462 | **0.364** | **0.408** | 0.524 | 0.455 | **0.371** | **0.406** |
| Traffic | 96 | 0.406 | 0.285 | **0.398** | **0.275** | **0.424** | **0.288** | 0.425 | 0.289 | 0.398 | 0.275 | **0.389** | **0.267** | 0.402 | 0.264 | **0.399** | **0.257** | **0.430** | 0.269 | 0.434 | **0.268** |
| | 192 | **0.421** | **0.289** | 0.425 | 0.290 | **0.433** | 0.296 | 0.434 | **0.296** | 0.417 | 0.286 | **0.407** | **0.277** | 0.416 | 0.271 | **0.416** | **0.265** | **0.435** | 0.302 | 0.444 | **0.272** |
| | 336 | **0.435** | 0.296 | 0.437 | 0.296 | **0.445** | 0.304 | 0.446 | **0.303** | 0.431 | 0.294 | **0.422** | **0.282** | 0.429 | 0.276 | **0.409** | **0.265** | 0.447 | 0.308 | 0.453 | **0.277** |
| | 720 | **0.465** | **0.309** | 0.472 | 0.316 | 0.474 | 0.322 | **0.474** | **0.320** | 0.463 | 0.312 | **0.456** | **0.301** | 0.456 | 0.293 | **0.447** | **0.281** | 0.476 | 0.327 | 0.477 | **0.293** |
| | Avg | **0.432** | 0.295 | 0.433 | **0.294** | **0.444** | 0.302 | 0.445 | **0.302** | 0.427 | 0.292 | **0.418** | **0.282** | 0.426 | 0.276 | **0.425** | **0.267** | **0.445** | 0.308 | 0.452 | **0.278** |
| Weather | 96 | 0.148 | 0.188 | **0.147** | **0.187** | 0.176 | 0.214 | **0.172** | **0.211** | 0.162 | 0.240 | **0.152** | **0.191** | 0.155 | 0.189 | **0.151** | **0.186** | 0.178 | 0.215 | **0.178** | **0.213** |
| | 192 | 0.193 | 0.232 | **0.189** | **0.227** | 0.217 | 0.251 | **0.217** | 0.251 | 0.213 | 0.249 | **0.195** | **0.238** | 0.194 | 0.230 | 0.195 | **0.230** | 0.218 | 0.252 | **0.217** | **0.252** |
| | 336 | 0.247 | 0.274 | **0.242** | **0.270** | 0.265 | 0.288 | **0.263** | **0.286** | 0.263 | 0.287 | **0.244** | **0.279** | 0.256 | 0.279 | **0.246** | **0.271** | 0.261 | 0.291 | **0.261** | 0.292 |
| | 720 | **0.318** | **0.323** | 0.325 | 0.327 | 0.333 | 0.334 | **0.332** | **0.334** | 0.339 | 0.340 | **0.315** | 0.338 | 0.327 | 0.327 | **0.327** | **0.326** | 0.326 | 0.354 | **0.322** | **0.344** |
| | Avg | 0.226 | 0.254 | **0.226** | **0.253** | 0.248 | 0.272 | **0.246** | **0.271** | 0.244 | 0.279 | **0.226** | **0.262** | 0.233 | 0.256 | **0.230** | **0.253** | 0.246 | 0.278 | **0.244** | **0.275** |
| Solar-Energy | 96 | 0.234 | 0.263 | **0.222** | 0.264 | 0.224 | 0.222 | **0.220** | **0.222** | 0.188 | 0.210 | **0.174** | **0.207** | 0.204 | 0.229 | **0.186** | **0.217** | 0.223 | 0.293 | 0.234 | **0.214** |
| | 192 | 0.266 | 0.287 | **0.251** | **0.278** | 0.256 | 0.242 | **0.248** | **0.239** | 0.213 | 0.226 | **0.204** | 0.233 | 0.211 | 0.225 | **0.201** | **0.224** | **0.251** | 0.311 | 0.266 | **0.234** |
| | 336 | 0.289 | 0.304 | **0.240** | **0.263** | 0.279 | 0.257 | **0.272** | **0.254** | 0.228 | 0.235 | **0.200** | **0.233** | 0.217 | **0.236** | 0.212 | 0.238 | **0.272** | 0.327 | 0.289 | **0.248** |
| | 720 | 0.290 | 0.315 | **0.254** | **0.264** | 0.280 | 0.260 | **0.261** | **0.250** | 0.233 | 0.237 | **0.204** | **0.227** | 0.220 | 0.236 | 0.226 | 0.237 | **0.274** | 0.330 | 0.288 | **0.248** |
| | Avg | 0.270 | 0.292 | **0.242** | **0.267** | 0.260 | 0.245 | **0.250** | **0.241** | 0.215 | 0.227 | **0.196** | **0.225** | 0.213 | 0.231 | **0.206** | **0.229** | 0.255 | 0.315 | 0.269 | **0.236** |
| Wind | 96 | 0.750 | 0.660 | **0.734** | **0.651** | 0.720 | 0.644 | **0.720** | **0.644** | 0.741 | 0.664 | **0.703** | **0.640** | 0.733 | 0.651 | **0.720** | **0.646** | **0.709** | 0.646 | 0.710 | **0.646** |
| | 192 | 0.770 | 0.681 | **0.759** | **0.669** | 0.759 | 0.669 | **0.759** | **0.669** | 0.779 | 0.685 | **0.735** | **0.663** | 0.758 | 0.677 | **0.758** | **0.671** | 0.743 | 0.672 | 0.744 | **0.672** |
| | 336 | 0.793 | 0.695 | **0.780** | **0.683** | 0.780 | 0.683 | **0.780** | **0.683** | 0.789 | 0.692 | **0.744** | **0.672** | 0.800 | 0.692 | **0.771** | **0.681** | 0.762 | 0.685 | **0.762** | **0.685** |
| | 720 | 0.818 | 0.711 | **0.804** | **0.703** | 0.811 | 0.704 | **0.810** | **0.704** | 0.812 | 0.712 | **0.745** | **0.679** | 0.811 | 0.706 | **0.794** | **0.700** | 0.781 | 0.703 | 0.782 | **0.703** |
| | Avg | 0.783 | 0.687 | **0.769** | **0.676** | 0.768 | 0.675 | **0.767** | **0.675** | 0.780 | 0.689 | **0.732** | **0.663** | 0.776 | 0.681 | **0.761** | **0.675** | 0.749 | 0.676 | 0.750 | **0.676** |

**Results under hyperparameter search.** In addition to the fixed-setting comparison (Table 2), we also report results under a hyperparameter search setup (Table 10). Following Appendix A.3, the lookback window is selected from $\{96, 192, 320, 512\}$ for each model, and the best configuration is reported. This ensures that the performance reflects each model's optimal capacity rather than being restricted by a fixed input length. As shown in Table 10, our MSAR consistently achieves superior results even when all baselines are tuned to their best configurations, demonstrating that the gains of MSAR are not tied to a particular hyperparameter choice but stem from its scale-aligned autoregressive design.

**Short-term forecasting results.** Table 11 reports short-term forecasting results on the PEMS datasets with a prediction length of 12. We evaluate three common metrics—MAE, MAPE, and RMSE—to provide a comprehensive view of performance. Across all four datasets (PEMS03, PEMS04, PEMS07, and PEMS08) and five representative backbones, incorporating MSAR consistently improves forecasting accuracy. Notably, the gains are most pronounced on PEMS04 and PEMS08, where error reductions are substantial across all metrics. These results confirm that the proposed scale-wise autoregressive framework is not only effective for long-term forecasting (as

Table 9: Full results of the long-term forecasting task. We compare extensive competitive models under different prediction lengths following the setting of TimesNet (Wu et al., 2022). The input sequence length is set to 336 for all baselines. *Avg* means the average results from all four prediction lengths.

| Models | | MSAR (Ours) | | SimpleTM (2025) | | DUET (2025) | | xPatch (2025) | | PatchMLP (2025) | | iTransformer (2024a) | | TimeMixer (2024a) | | PatchTST (2023) | | LightTS (2023) | | DLinear (2023) | | FreTS (2023) | |
|---|---|---|---|---|---|---|---|---|---|---|---|---|---|---|---|---|---|---|---|---|---|---|---|
| Metric | | MSE | MAE | MSE | MAE | MSE | MAE | MSE | MAE | MSE | MAE | MSE | MAE | MSE | MAE | MSE | MAE | MSE | MAE | MSE | MAE | MSE | MAE |
| ECL | 96 | 0.132 | 0.226 | 0.134 | 0.226 | 0.149 | 0.251 | 0.143 | 0.241 | 0.165 | 0.271 | 0.135 | 0.229 | 0.141 | 0.238 | 0.138 | 0.231 | 0.171 | 0.285 | 0.143 | 0.242 | 0.144 | 0.245 |
| | 192 | 0.149 | 0.241 | 0.154 | 0.245 | 0.163 | 0.264 | 0.157 | 0.252 | 0.183 | 0.288 | 0.153 | 0.245 | 0.158 | 0.254 | 0.154 | 0.246 | 0.192 | 0.305 | 0.157 | 0.255 | 0.156 | 0.253 |
| | 336 | 0.164 | 0.257 | 0.175 | 0.268 | 0.181 | 0.281 | 0.172 | 0.266 | 0.201 | 0.304 | 0.169 | 0.261 | 0.175 | 0.270 | 0.171 | 0.261 | 0.214 | 0.324 | 0.172 | 0.272 | 0.173 | 0.272 |
| | 720 | 0.196 | 0.285 | 0.191 | 0.278 | 0.220 | 0.311 | 0.209 | 0.297 | 0.250 | 0.343 | 0.209 | 0.294 | 0.217 | 0.307 | 0.210 | 0.294 | 0.251 | 0.349 | 0.207 | 0.305 | 0.210 | 0.308 |
| | Avg | 0.160 | 0.252 | 0.164 | 0.254 | 0.178 | 0.277 | 0.170 | 0.264 | 0.200 | 0.301 | 0.167 | 0.257 | 0.173 | 0.267 | 0.168 | 0.258 | 0.207 | 0.316 | 0.170 | 0.269 | 0.171 | 0.270 |
| ETTm1 | 96 | 0.280 | 0.327 | 0.291 | 0.337 | 0.292 | 0.335 | 0.293 | 0.347 | 0.305 | 0.357 | 0.318 | 0.355 | 0.307 | 0.352 | 0.317 | 0.343 | 0.313 | 0.363 | 0.293 | 0.333 | 0.308 | 0.353 |
| | 192 | 0.324 | 0.355 | 0.330 | 0.363 | 0.332 | 0.359 | 0.332 | 0.371 | 0.351 | 0.384 | 0.370 | 0.383 | 0.332 | 0.373 | 0.335 | 0.361 | 0.350 | 0.387 | 0.332 | 0.355 | 0.346 | 0.377 |
| | 336 | 0.357 | 0.376 | 0.369 | 0.384 | 0.365 | 0.378 | 0.364 | 0.395 | 0.387 | 0.403 | 0.400 | 0.403 | 0.362 | 0.390 | 0.381 | 0.384 | 0.399 | 0.418 | 0.366 | 0.376 | 0.382 | 0.399 |
| | 720 | 0.414 | 0.409 | 0.425 | 0.416 | 0.419 | 0.409 | 0.428 | 0.432 | 0.454 | 0.441 | 0.462 | 0.434 | 0.422 | 0.426 | 0.451 | 0.427 | 0.489 | 0.479 | 0.426 | 0.417 | 0.440 | 0.433 |
| | Avg | 0.344 | 0.367 | 0.354 | 0.375 | 0.352 | 0.370 | 0.354 | 0.386 | 0.374 | 0.396 | 0.388 | 0.394 | 0.356 | 0.385 | 0.371 | 0.379 | 0.388 | 0.412 | 0.354 | 0.370 | 0.369 | 0.391 |
| ETTm2 | 96 | 0.163 | 0.246 | 0.177 | 0.255 | 0.163 | 0.248 | 0.168 | 0.259 | 0.179 | 0.265 | 0.179 | 0.266 | 0.169 | 0.258 | 0.173 | 0.258 | 0.187 | 0.286 | 0.165 | 0.248 | 0.175 | 0.263 |
| | 192 | 0.219 | 0.285 | 0.251 | 0.302 | 0.219 | 0.285 | 0.226 | 0.299 | 0.232 | 0.301 | 0.254 | 0.323 | 0.240 | 0.308 | 0.219 | 0.288 | 0.311 | 0.384 | 0.222 | 0.288 | 0.248 | 0.314 |
| | 336 | 0.272 | 0.319 | 0.305 | 0.338 | 0.273 | 0.322 | 0.281 | 0.337 | 0.285 | 0.334 | 0.300 | 0.350 | 0.282 | 0.334 | 0.279 | 0.327 | 0.399 | 0.418 | 0.276 | 0.327 | 0.321 | 0.364 |
| | 720 | 0.355 | 0.378 | 0.399 | 0.394 | 0.365 | 0.379 | 0.372 | 0.391 | 0.380 | 0.394 | 0.389 | 0.399 | 0.369 | 0.391 | 0.361 | 0.381 | 0.489 | 0.479 | 0.375 | 0.393 | 0.393 | 0.414 |
| | Avg | 0.252 | 0.307 | 0.283 | 0.322 | 0.255 | 0.308 | 0.255 | 0.322 | 0.269 | 0.324 | 0.280 | 0.335 | 0.265 | 0.323 | 0.258 | 0.314 | 0.347 | 0.392 | 0.260 | 0.314 | 0.284 | 0.339 |
| ETTh1 | 96 | 0.358 | 0.381 | 0.379 | 0.401 | 0.359 | 0.383 | 0.380 | 0.403 | 0.411 | 0.426 | 0.409 | 0.415 | 0.376 | 0.398 | 0.383 | 0.403 | 0.432 | 0.443 | 0.365 | 0.383 | 0.400 | 0.419 |
| | 192 | 0.397 | 0.406 | 0.423 | 0.430 | 0.398 | 0.406 | 0.425 | 0.430 | 0.443 | 0.445 | 0.507 | 0.462 | 0.412 | 0.420 | 0.409 | 0.424 | 0.474 | 0.472 | 0.417 | 0.426 | 0.436 | 0.441 |
| | 336 | 0.422 | 0.423 | 0.440 | 0.443 | 0.426 | 0.422 | 0.444 | 0.439 | 0.468 | 0.461 | 0.459 | 0.451 | 0.445 | 0.446 | 0.440 | 0.446 | 0.515 | 0.501 | 0.443 | 0.436 | 0.471 | 0.466 |
| | 720 | 0.433 | 0.452 | 0.455 | 0.475 | 0.430 | 0.450 | 0.534 | 0.509 | 0.525 | 0.508 | 0.526 | 0.506 | 0.488 | 0.489 | 0.481 | 0.482 | 0.592 | 0.567 | 0.456 | 0.473 | 0.537 | 0.532 |
| | Avg | 0.403 | 0.415 | 0.424 | 0.437 | 0.403 | 0.415 | 0.446 | 0.445 | 0.462 | 0.460 | 0.475 | 0.459 | 0.430 | 0.438 | 0.428 | 0.439 | 0.503 | 0.496 | 0.420 | 0.429 | 0.461 | 0.465 |
| ETTh2 | 96 | 0.272 | 0.330 | 0.295 | 0.342 | 0.273 | 0.331 | 0.280 | 0.343 | 0.311 | 0.371 | 0.343 | 0.373 | 0.285 | 0.346 | 0.292 | 0.346 | 0.324 | 0.383 | 0.278 | 0.333 | 0.321 | 0.380 |
| | 192 | 0.338 | 0.375 | 0.361 | 0.388 | 0.347 | 0.382 | 0.345 | 0.386 | 0.381 | 0.413 | 0.410 | 0.413 | 0.358 | 0.391 | 0.354 | 0.385 | 0.438 | 0.455 | 0.350 | 0.382 | 0.396 | 0.429 |
| | 336 | 0.365 | 0.400 | 0.426 | 0.428 | 0.394 | 0.415 | 0.382 | 0.415 | 0.401 | 0.427 | 0.419 | 0.434 | 0.380 | 0.409 | 0.383 | 0.407 | 0.496 | 0.490 | 0.403 | 0.424 | 0.497 | 0.496 |
| | 720 | 0.387 | 0.420 | 0.427 | 0.446 | 0.405 | 0.433 | 0.408 | 0.443 | 0.426 | 0.451 | 0.448 | 0.461 | 0.410 | 0.439 | 0.587 | 0.549 | 1.151 | 0.749 | 0.556 | 0.518 | 0.766 | 0.628 |
| | Avg | 0.341 | 0.381 | 0.377 | 0.401 | 0.355 | 0.390 | 0.354 | 0.397 | 0.380 | 0.416 | 0.405 | 0.420 | 0.358 | 0.396 | 0.404 | 0.422 | 0.602 | 0.519 | 0.397 | 0.414 | 0.495 | 0.483 |
| Exchange | 96 | 0.088 | 0.206 | 0.094 | 0.216 | 0.092 | 0.213 | 0.256 | 0.367 | 0.094 | 0.218 | 0.109 | 0.239 | 0.088 | 0.218 | 0.098 | 0.221 | 0.148 | 0.278 | 0.082 | 0.201 | 0.197 | 0.323 |
| | 192 | 0.154 | 0.283 | 0.190 | 0.313 | 0.189 | 0.307 | 0.470 | 0.509 | 0.184 | 0.307 | 0.266 | 0.377 | 0.176 | 0.315 | 0.210 | 0.329 | 0.271 | 0.315 | 0.174 | 0.299 | 0.300 | 0.369 |
| | 336 | 0.276 | 0.392 | 0.331 | 0.417 | 0.392 | 0.456 | 1.268 | 0.883 | 0.349 | 0.431 | 0.396 | 0.467 | 0.313 | 0.427 | 0.468 | 0.506 | 0.460 | 0.427 | 0.347 | 0.427 | 0.509 | 0.524 |
| | 720 | 0.791 | 0.659 | 1.138 | 0.784 | 0.901 | 0.714 | 1.767 | 1.068 | 0.852 | 0.698 | 0.841 | 0.691 | 0.839 | 0.695 | 1.082 | 0.793 | 1.195 | 0.695 | 1.493 | 0.893 | 1.447 | 0.941 |
| | Avg | 0.327 | 0.385 | 0.438 | 0.432 | 0.394 | 0.422 | 0.940 | 0.707 | 0.370 | 0.413 | 0.403 | 0.444 | 0.354 | 0.414 | 0.465 | 0.462 | 0.518 | 0.429 | 0.524 | 0.455 | 0.613 | 0.539 |
| Traffic | 96 | 0.389 | 0.257 | 0.406 | 0.285 | 0.424 | 0.288 | 0.413 | 0.297 | 0.486 | 0.361 | 0.398 | 0.275 | 0.421 | 0.297 | 0.402 | 0.264 | 0.493 | 0.367 | 0.430 | 0.269 | 0.417 | 0.301 |
| | 192 | 0.407 | 0.265 | 0.421 | 0.289 | 0.433 | 0.296 | 0.425 | 0.298 | 0.515 | 0.378 | 0.417 | 0.286 | 0.441 | 0.311 | 0.416 | 0.271 | 0.511 | 0.373 | 0.435 | 0.302 | 0.447 | 0.299 |
| | 336 | 0.409 | 0.265 | 0.435 | 0.296 | 0.445 | 0.304 | 0.430 | 0.297 | 0.533 | 0.386 | 0.431 | 0.294 | 0.425 | 0.317 | 0.429 | 0.276 | 0.521 | 0.378 | 0.447 | 0.308 | 0.468 | 0.311 |
| | 720 | 0.447 | 0.281 | 0.465 | 0.309 | 0.474 | 0.322 | 0.450 | 0.306 | 0.564 | 0.401 | 0.463 | 0.312 | 0.489 | 0.340 | 0.456 | 0.293 | 0.504 | 0.360 | 0.476 | 0.327 | 0.512 | 0.340 |
| | Avg | 0.413 | 0.267 | 0.432 | 0.295 | 0.444 | 0.302 | 0.429 | 0.299 | 0.524 | 0.382 | 0.427 | 0.292 | 0.444 | 0.316 | 0.426 | 0.276 | 0.507 | 0.369 | 0.447 | 0.301 | 0.461 | 0.313 |
| Weather | 96 | 0.147 | 0.186 | 0.148 | 0.188 | 0.176 | 0.214 | 0.148 | 0.196 | 0.153 | 0.206 | 0.162 | 0.204 | 0.150 | 0.199 | 0.155 | 0.189 | 0.152 | 0.212 | 0.178 | 0.215 | 0.156 | 0.213 |
| | 192 | 0.189 | 0.227 | 0.193 | 0.232 | 0.217 | 0.251 | 0.190 | 0.238 | 0.197 | 0.245 | 0.213 | 0.249 | 0.193 | 0.244 | 0.194 | 0.230 | 0.197 | 0.255 | 0.218 | 0.252 | 0.197 | 0.253 |
| | 336 | 0.242 | 0.270 | 0.247 | 0.274 | 0.265 | 0.288 | 0.241 | 0.280 | 0.247 | 0.282 | 0.263 | 0.287 | 0.244 | 0.283 | 0.256 | 0.279 | 0.249 | 0.299 | 0.261 | 0.291 | 0.246 | 0.293 |
| | 720 | 0.315 | 0.326 | 0.318 | 0.323 | 0.333 | 0.334 | 0.318 | 0.333 | 0.324 | 0.334 | 0.339 | 0.340 | 0.319 | 0.336 | 0.327 | 0.327 | 0.323 | 0.357 | 0.326 | 0.354 | 0.319 | 0.347 |
| | Avg | 0.223 | 0.252 | 0.226 | 0.254 | 0.248 | 0.272 | 0.224 | 0.262 | 0.230 | 0.267 | 0.244 | 0.270 | 0.226 | 0.266 | 0.233 | 0.256 | 0.230 | 0.281 | 0.246 | 0.278 | 0.229 | 0.276 |
| Solar-Energy | 96 | 0.174 | 0.207 | 0.234 | 0.263 | 0.224 | 0.222 | 0.186 | 0.240 | 0.211 | 0.280 | 0.188 | 0.210 | 0.199 | 0.257 | 0.204 | 0.229 | 0.196 | 0.267 | 0.223 | 0.293 | 0.190 | 0.248 |
| | 192 | 0.201 | 0.224 | 0.266 | 0.287 | 0.256 | 0.242 | 0.199 | 0.254 | 0.245 | 0.285 | 0.213 | 0.226 | 0.221 | 0.274 | 0.211 | 0.225 | 0.221 | 0.280 | 0.251 | 0.311 | 0.209 | 0.266 |
| | 336 | 0.200 | 0.233 | 0.289 | 0.304 | 0.279 | 0.257 | 0.205 | 0.257 | 0.271 | 0.302 | 0.228 | 0.235 | 0.230 | 0.284 | 0.217 | 0.236 | 0.229 | 0.299 | 0.272 | 0.327 | 0.222 | 0.274 |
| | 720 | 0.204 | 0.227 | 0.290 | 0.315 | 0.280 | 0.260 | 0.209 | 0.261 | 0.272 | 0.303 | 0.233 | 0.237 | 0.224 | 0.290 | 0.220 | 0.236 | 0.232 | 0.304 | 0.274 | 0.330 | 0.228 | 0.278 |
| | Avg | 0.195 | 0.223 | 0.270 | 0.292 | 0.260 | 0.245 | 0.200 | 0.253 | 0.250 | 0.292 | 0.215 | 0.227 | 0.218 | 0.276 | 0.213 | 0.231 | 0.217 | 0.288 | 0.255 | 0.315 | 0.212 | 0.267 |
| Wind | 96 | 0.703 | 0.640 | 0.750 | 0.660 | 0.720 | 0.644 | 0.717 | 0.653 | 0.740 | 0.668 | 0.741 | 0.664 | 0.724 | 0.658 | 0.733 | 0.651 | 0.695 | 0.645 | 0.709 | 0.646 | 0.711 | 0.656 |
| | 192 | 0.735 | 0.663 | 0.770 | 0.681 | 0.759 | 0.669 | 0.755 | 0.677 | 0.766 | 0.685 | 0.779 | 0.685 | 0.751 | 0.678 | 0.758 | 0.677 | 0.726 | 0.669 | 0.743 | 0.672 | 0.737 | 0.675 |
| | 336 | 0.744 | 0.672 | 0.793 | 0.695 | 0.780 | 0.683 | 0.774 | 0.689 | 0.783 | 0.695 | 0.789 | 0.696 | 0.777 | 0.692 | 0.800 | 0.692 | 0.733 | 0.676 | 0.762 | 0.685 | 0.752 | 0.685 |
| | 720 | 0.745 | 0.679 | 0.818 | 0.711 | 0.811 | 0.704 | 0.803 | 0.710 | 0.799 | 0.709 | 0.812 | 0.712 | 0.796 | 0.710 | 0.811 | 0.706 | 0.749 | 0.689 | 0.781 | 0.703 | 0.769 | 0.702 |
| | Avg | 0.732 | 0.663 | 0.783 | 0.687 | 0.768 | 0.675 | 0.762 | 0.682 | 0.772 | 0.689 | 0.780 | 0.689 | 0.762 | 0.684 | 0.776 | 0.681 | 0.726 | 0.670 | 0.749 | 0.676 | 0.742 | 0.679 |
| 1st Count | | 30 | 29 | 1 | 4 | 1 | 5 | 2 | 0 | 0 | 0 | 1 | 0 | 3 | 0 | 1 | 0 | 3 | 0 | 1 | 3 | 0 | 0 |

Table 10: Results under the hyperparameter search setting described in Appendix Section A.3. The lookback window is selected from $\{96, 192, 320, 512\}$, and the boldres configuration is reported for each model. This setup ensures that the comparison reflects each model's optimal performance rather than a fixed setting constraint.

| Models | | MSAR Ours | | SimpleTM (2025) | | DUET (2025) | | iTransformer (2024a) | | PatchTST (2023) | | DLinear (2023) | |
|---|---|---|---|---|---|---|---|---|---|---|---|---|---|
| Metric | | MSE | MAE | MSE | MAE | MSE | MAE | MSE | MAE | MSE | MAE | MSE | MAE |
| ETTm1 | 96 | 0.289 | 0.335 | 0.280 | 0.337 | 0.300 | 0.345 | 0.299 | 0.348 | 0.289 | 0.343 | 0.299 | 0.343 |
| | 192 | 0.325 | 0.355 | 0.326 | 0.365 | 0.335 | 0.366 | 0.334 | 0.373 | 0.332 | 0.367 | 0.336 | 0.367 |
| | 336 | 0.355 | 0.376 | 0.362 | 0.387 | 0.364 | 0.383 | 0.370 | 0.394 | 0.362 | 0.389 | 0.367 | 0.385 |
| | 720 | 0.410 | 0.406 | 0.417 | 0.422 | 0.417 | 0.413 | 0.422 | 0.426 | 0.408 | 0.420 | 0.420 | 0.418 |
| | Avg. | 0.345 | 0.368 | 0.346 | 0.378 | 0.354 | 0.377 | 0.356 | 0.385 | 0.348 | 0.380 | 0.356 | 0.378 |
| ETTm2 | 96 | 0.160 | 0.246 | 0.168 | 0.252 | 0.162 | 0.253 | 0.180 | 0.263 | 0.165 | 0.257 | 0.165 | 0.260 |
| | 192 | 0.216 | 0.284 | 0.223 | 0.297 | 0.217 | 0.290 | 0.236 | 0.307 | 0.223 | 0.294 | 0.219 | 0.297 |
| | 336 | 0.269 | 0.321 | 0.273 | 0.328 | 0.270 | 0.325 | 0.287 | 0.340 | 0.272 | 0.333 | 0.290 | 0.345 |
| | 720 | 0.359 | 0.377 | 0.364 | 0.386 | 0.361 | 0.384 | 0.364 | 0.391 | 0.363 | 0.386 | 0.375 | 0.403 |
| | Avg. | 0.251 | 0.307 | 0.257 | 0.316 | 0.253 | 0.313 | 0.267 | 0.325 | 0.256 | 0.318 | 0.262 | 0.326 |
| ETTh1 | 96 | 0.356 | 0.381 | 0.375 | 0.394 | 0.364 | 0.392 | 0.380 | 0.401 | 0.370 | 0.397 | 0.368 | 0.393 |
| | 192 | 0.395 | 0.407 | 0.412 | 0.429 | 0.397 | 0.413 | 0.423 | 0.431 | 0.404 | 0.423 | 0.400 | 0.417 |
| | 336 | 0.426 | 0.424 | 0.429 | 0.443 | 0.422 | 0.432 | 0.437 | 0.450 | 0.423 | 0.437 | 0.430 | 0.441 |
| | 720 | 0.428 | 0.449 | 0.447 | 0.472 | 0.443 | 0.463 | 0.460 | 0.471 | 0.444 | 0.464 | 0.477 | 0.497 |
| | Avg. | 0.401 | 0.417 | 0.416 | 0.434 | 0.406 | 0.425 | 0.425 | 0.438 | 0.410 | 0.430 | 0.419 | 0.437 |
| ETTh2 | 96 | 0.267 | 0.328 | 0.291 | 0.339 | 0.265 | 0.334 | 0.294 | 0.345 | 0.290 | 0.342 | 0.286 | 0.351 |
| | 192 | 0.331 | 0.362 | 0.349 | 0.390 | 0.324 | 0.373 | 0.358 | 0.395 | 0.354 | 0.397 | 0.352 | 0.394 |
| | 336 | 0.364 | 0.399 | 0.381 | 0.413 | 0.352 | 0.399 | 0.385 | 0.415 | 0.384 | 0.413 | 0.439 | 0.456 |
| | 720 | 0.397 | 0.430 | 0.410 | 0.442 | 0.393 | 0.432 | 0.412 | 0.436 | 0.408 | 0.442 | 0.570 | 0.530 |
| | Avg. | 0.340 | 0.380 | 0.358 | 0.396 | 0.334 | 0.384 | 0.362 | 0.398 | 0.359 | 0.399 | 0.412 | 0.433 |
| Weather | 96 | 0.145 | 0.189 | 0.143 | 0.195 | 0.168 | 0.221 | 0.155 | 0.206 | 0.145 | 0.194 | 0.170 | 0.229 |
| | 192 | 0.188 | 0.230 | 0.188 | 0.236 | 0.212 | 0.258 | 0.199 | 0.249 | 0.191 | 0.237 | 0.212 | 0.268 |
| | 336 | 0.257 | 0.284 | 0.238 | 0.276 | 0.258 | 0.292 | 0.247 | 0.284 | 0.243 | 0.279 | 0.258 | 0.307 |
| | 720 | 0.323 | 0.332 | 0.312 | 0.327 | 0.324 | 0.338 | 0.318 | 0.336 | 0.313 | 0.330 | 0.321 | 0.359 |
| | Avg. | 0.228 | 0.259 | 0.220 | 0.259 | 0.240 | 0.277 | 0.230 | 0.269 | 0.223 | 0.260 | 0.240 | 0.291 |
| 1st Count | | 37 | | 6 | | 6 | | 0 | | 1 | | 0 | |

shown in the main paper), but also transfers well to high-frequency short-term forecasting tasks, highlighting its generality and robustness.

## C  EFFICIENCY STUDY

To assess the inference efficiency of our MSAR framework, we report the wall-clock runtime of four backbone architectures—DLinear [22], TimesNet [21], PatchTST [10], and iTransformer [8]—both with and without MSAR integration. Additionally, we compare against SegRNN [7], a representative autoregressive baseline that sequentially generates future time steps.

Figure 6 compares the inference time of baseline models with their MSAR-enhanced counterparts under identical input-output settings (input length = 960, prediction length = 336, batch size = 32). For most backbones, including TimesNet [21] and PatchTST [10], the hierarchical design of MSAR improves both prediction accuracy and runtime by reducing redundant computation through coarse-scale anchoring. However, we observe an increase in inference time when integrating MSAR with iTransformer [8]. This is expected, as iTransformer encodes inputs via a global linear projection into the latent space without explicit dependency on sequence length. The hierarchical multi-scale processing in MSAR introduces additional overhead in such architectures, where lookback windows do not otherwise impact runtime.

Table 11: Short-term forecasting results on PEMS datasets (prediction length = 12). Lower MAE, MAPE, and MSE indicate better performance.

| Dataset | Metric | SimpleTM | | DUET | | iTransformer | | PatchTST | | DLinear | |
|---|---|---|---|---|---|---|---|---|---|---|---|
| | | Base | +Ours | Base | +Ours | Base | +Ours | Base | +Ours | Base | +Ours |
| PEMS03 | MAE | 19.147 | **15.627** | 18.853 | **18.402** | 15.870 | **15.308** | 16.974 | **15.209** | 17.411 | **16.743** |
| | MAPE | 19.527 | **15.796** | 22.031 | **20.698** | 15.945 | **15.459** | 19.376 | **15.905** | 15.876 | **15.509** |
| | RMSE | 30.307 | **24.933** | 29.607 | **29.240** | 25.563 | **24.648** | 26.517 | **24.310** | 29.619 | **27.909** |
| PEMS04 | MAE | 27.192 | **21.651** | 24.581 | 24.693 | 22.033 | **20.617** | 23.005 | **21.022** | 24.359 | **22.546** |
| | MAPE | 18.064 | **13.462** | 18.659 | **18.202** | 14.037 | **13.019** | 15.779 | **13.846** | 14.616 | **13.538** |
| | RMSE | 41.719 | **34.425** | 37.486 | 37.614 | 34.834 | **33.023** | 35.616 | **33.300** | 38.966 | **36.244** |
| PEMS07 | MAE | 26.999 | **22.770** | 26.862 | **26.577** | 23.064 | **21.863** | 24.010 | **21.911** | 25.519 | **24.655** |
| | MAPE | 12.158 | **9.675** | 13.401 | **13.258** | 10.119 | **9.544** | 10.821 | **9.311** | 10.869 | **10.384** |
| | RMSE | 41.257 | **35.833** | 40.314 | **40.114** | 36.330 | **34.762** | 36.711 | **33.92** | 40.313 | **39.109** |
| PEMS08 | MAE | 22.261 | **16.412** | 19.922 | **19.976** | 17.350 | **15.851** | 18.591 | **16.057** | 20.353 | **18.090** |
| | MAPE | 14.124 | **10.086** | 13.631 | **13.459** | 10.963 | **9.990** | 12.569 | **10.419** | 12.376 | **11.047** |
| | RMSE | 34.455 | **26.059** | 30.199 | **30.720** | 27.459 | **25.312** | 28.172 | **25.267** | 32.658 | **29.043** |

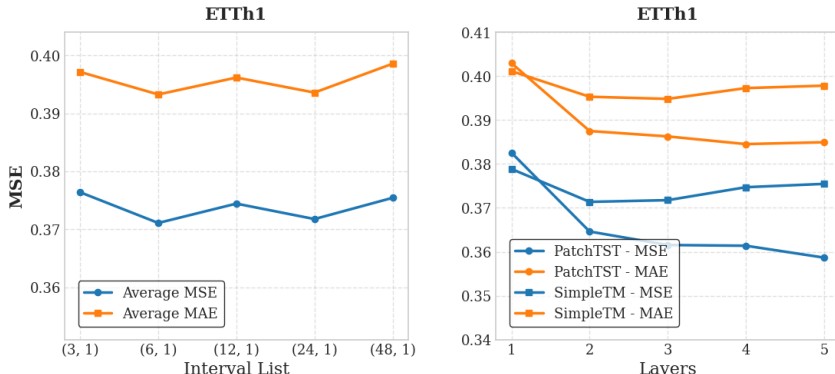

Figure 5: Sensitivity analysis of MSAR on the ETTh1 dataset. **Top:** Effect of different interval lists, averaged over five backbone models. **Bottom:** Effect of the number of MSAR layers for PatchTST and SimpleTM.

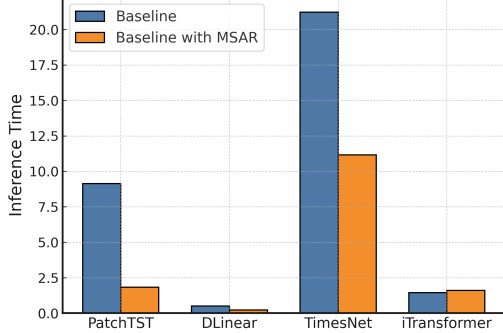

| Model | H | Inference Time (s) |
|---|---|---|
| SegRNN[6] | 336 | 7.23 |
| SegRNN[12] | 336 | 4.06 |
| SegRNN[6] | 720 | 8.53 |
| SegRNN[12] | 720 | 5.01 |
| PatchTST* | 336 | 2.13 |
| PatchTST* | 720 | 2.30 |
| iTransformer* | 336 | 1.84 |
| iTransformer* | 720 | 1.94 |
| DLinear* | 336 | 0.31 |
| DLinear* | 720 | 0.36 |

Figure 6: Inference efficiency of baseline models vs. MSAR-enhanced models (batch size 32, input=960, pred=336).

Table 12: Inference time (in seconds) for different models and prediction lengths **H**. * indicates MSAR-enhanced models. SegRNN[6] and SegRNN[12] denote segment length.

Table 12 collectively demonstrates the inference efficiency of MSAR-enhanced models compared to autoregressive baselines on the ETTh1 dataset. It shows that MSAR consistently achieves significant inference speedups over SegRNN [7], as it avoids full-step autoregression by leveraging parallel decoding within each scale. MSAR achieves favorable efficiency-accuracy trade-offs across backbones, demonstrating its generalizability and practical utility for scalable time series forecasting.

## D   ADDITIONAL PREDICTION RESULTS

In this section, we provide additional qualitative results on the ETTh1 dataset. All experiments are conducted with a single-layer encoder (`Layer=1`), two attention heads (`Head=2`), and an interval configuration of `interval_list = [3, 1]`. We evaluate the model under four input sequence length settings: `[32, 96]`, `[32, 192]`, `[32, 384]`, and `[32, 512]`. The visualizations below are generated using our proposed **DUET+MSAR** model, showing its prediction accuracy compared to the ground-truth time series.

The visualizations in Figure 7 are generated using our proposed **DUET+MSAR** model, illustrating its prediction accuracy compared to the ground-truth time series.

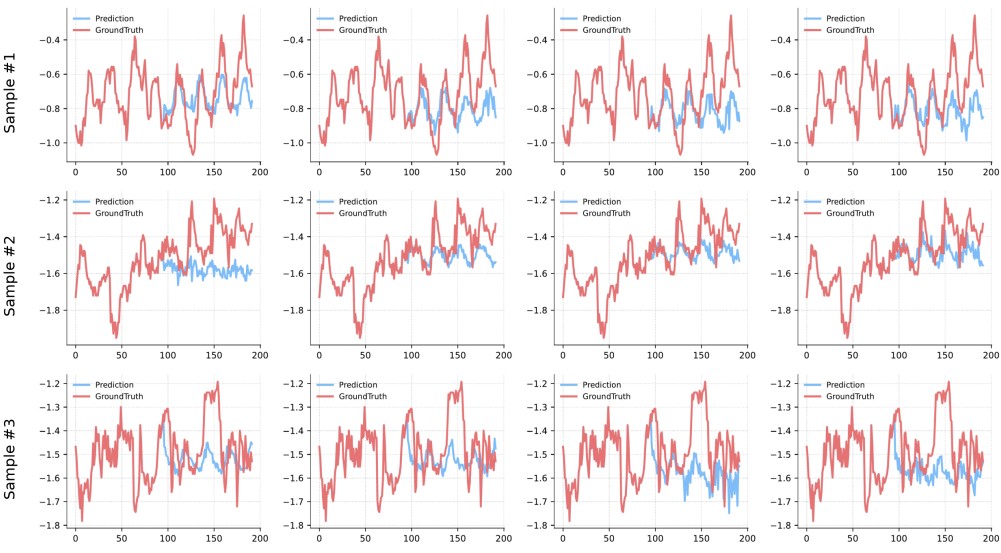

Figure 7: Visualization of model predictions and ground truth on the ETTh1 dataset. Results are produced by the **DUET+MSAR** model.

## E   THE USE OF LARGE LANGUAGE MODELS (LLMS)

In this work, we employed large language models (LLMs) solely as auxiliary assistants for data processing and manuscript preparation. Specifically, LLMs were used to facilitate tasks such as organizing experimental logs, generating LaTeX table skeletons, and providing language refinement suggestions. Importantly, LLMs were not involved in designing algorithms, building models, or analyzing experimental outcomes. All technical contributions, theoretical analyses, and empirical results presented in this paper are entirely the work of the authors.

