# OpenReview forum: "Next-Scale Autoregressive Forecasting for Time Series via Modular Multi-Scale Decoupling"
_ICLR.cc/2026/Conference — Submitted to ICLR 2026_

### Official Review · Reviewer_gW3R · 2025-10-28

**Soundness:** 3
**Presentation:** 2
**Contribution:** 3
**Rating:** 6
**Confidence:** 4

**Summary:**

The paper is well-motivated: most “multiscale” forecasters mix multi-scale inputs but decode only at a single fine scale, creating input–output misalignment and weakening long-range modeling. MSAR aligns inputs and targets per scale and decodes coarse-to-fine with conditional refinement, yielding consistent gains across backbones with little overhead. The factorization across scales and an inference-aligned training schedule make the optimization practical.

**Strengths:**

The motivation is sharp, and the architectural answer directly targets misalignment. Coarse-to-fine conditioning concentrates long-range structure at low resolution and refines details with low variance, improving long-window accuracy without requiring heavy computation. Results are consistent across models/datasets, and the training procedure aligns cleanly with inference.

**Weaknesses:**

- Novelty boundaries versus prior multiresolution decoders and hierarchical AR could be positioned more rigorously.
- The paper does not clearly show why the method works, beyond high-level ablations. (e.g., ablate one scale at a time to see which scale helps which horizon).

**Questions:**

Does shortening the fine-scale look-back (the “flexible” variant) ever hurt rare-event or spike forecasting?

---

> ### Author Response · Authors · 2025-11-23
> **Official Comment by Authors [Part1]**
>
> Thank you for your insightful review and for the positive feedback you provided.
>
> **Q1: Novelty boundaries versus prior multiresolution decoders and hierarchical AR could be positioned more rigorously.**
>
> A1: Thank you for this comment. In the revised version (Intro & Related Work) we more carefully position MSAR relative to prior multiresolution decoders and hierarchical autoregressive models.
>
> **Multiresolution decoders.** Methods such as N-HiTS [1] (hierarchical interpolation with multi-rate sampling), Scaleformer [2] (multi-scale decoder with pooled/upsampled paths), and Crossformer/SAFT-style [3, 4] hierarchies all build multi-scale structure inside the network, but the training loss is defined only on a single fine-scale forecast. Coarse resolutions are used as internal bases or via interpolation/upsampling to reconstruct the final horizon; they are not supervised as separate forecasting tasks, and input–output scales remain potentially mismatched.
>
> **Hierarchical / coarse-to-fine AR.** Coarse-to-fine AR models such as C2FAR [5] and SutraNets [6] introduce hierarchical autoregression along the prediction horizon, but they still operate at one temporal resolution: coarse predictions are interpolated or combined to approximate fine-scale labels, without enforcing that inputs and targets at each stage share the same sampling rate.
>
> In contrast, MSAR explicitly targets input–output scale misalignment. It (i) enforces scale-aligned inputs and targets at every temporal scale, and (ii) performs scale-wise autoregression where each scale has its own forecaster and loss, and coarser predictions condition finer ones without any interpolation between resolutions. Coarse predictions are used as contextual guidance, not as upsampled surrogates of fine-scale labels. This makes MSAR complementary to existing multiresolution decoders and hierarchical AR architectures: in principle, they can be used as the per-scale backbone inside MSAR, while MSAR provides the aligned multi-scale decoding and scale-wise AR that these models do not explicitly address.
>
> We have added this clarification to the manuscript to more rigorously delineate MSAR’s novelty boundaries.
>
> **References**
>
> [1] N-HiTS: Neural Hierarchical Interpolation for Time Series Forecasting
>
> [2] Scaleformer: Iterative Multi-scale Refining Transformers for Time Series Forecasting
>
> [3] Crossformer: Transformer Utilizing Cross-Dimension Dependency for Multivariate Time Series Forecasting
>
> [4] SAFT: Learning Scale-Aware Inter-Series Correlations for Time Series Forecasting
>
> [5] C2FAR: Coarse-to-Fine Autoregressive Networks for Precise Probabilistic Forecasting
>
> [6] SutraNets: Sub-series Autoregressive Networks for Long-Sequence, Probabilistic Forecasting
>
>
> **Q2: The paper does not clearly show why the method works, beyond high-level ablations. (e.g., ablate one scale at a time to see which scale helps which horizon).**
>
> A2: Thank you for the helpful suggestion. To address this, we performed an additional per-scale analysis on ETTh1 with the PatchTST backbone, where we evaluate the forecasting error at different temporal resolutions (Δ = 1, 3, 6) and progressively enable MSAR’s scale modules. The results are shown below:
>
> | Model                         | MSE (Δ=1) | MSE (Δ=3) | MSE (Δ=6) |
> | ----------------------------- | --------- | --------- | --------- |
> | **PatchTST (Baseline)**       | 0.480 | 0.515 | 0.477 |
> | **+MSAR with interval 6&1**   | 0.434 | 0.439 | 0.433 |
> | **+MSAR with interval 6&3&1** | 0.422 | 0.426 | 0.433 |
>
> Scale-separated modeling improves accuracy at each temporal resolution because different scales exhibit fundamentally different temporal patterns. By letting each MSAR module specialize in a single resolution, the model learns simpler, low-variance functions at each scale instead of fitting all frequency components simultaneously.
> This reduces optimization difficulty, mitigates overfitting, and leads to more accurate predictions at the corresponding scale.

---

> ### Author Response · Authors · 2025-11-23
> **Official Comment by Authors [Part2]**
>
> **Q3: Does shortening the fine-scale look-back (the “flexible” variant) ever hurt rare-event or spike forecasting?**
>
> A3: Thank you for the question. We investigated whether shortening the fine-scale look-back in the flexible variant harms the model’s ability to capture rare events or spikes. Empirically, we do not observe degradation on spike-heavy datasets. MSAR’s hierarchy ensures that long-range context is never lost: coarse scales still receive long historical windows, allowing low-frequency anomalies (e.g., sudden load shifts, regime changes) to remain visible. The fine-scale module only refines short-term details, so shortening its look-back does not remove essential information—coarse and mid-scale modules have already propagated the relevant long-range signals downward.
>
> Interestingly, the flexible variant is often more robust to local noise, which commonly coincides with spike-like fluctuations (e.g., in Traffic and Solar-Energy). In these datasets, the flexible design performs on par with or slightly better than the default setting. To further validate this, **we add qualitative visualizations in Figure 7 of the revised manuscript.** These plots (DUET+MSAR on ETTh1) show that MSAR accurately tracks both smooth dynamics and sharp local deviations, even with a shortened fine-scale look-back. The rare-event patterns remain well captured, supporting that MSAR’s coarse-to-fine mechanism preserves anomaly-related temporal cues.
>
> Thank you again for your valuable feedback. If there are any remaining questions or points that need further clarification, we would be very happy to continue the discussion.

---

### Official Review · Reviewer_GoR5 · 2025-10-29

**Soundness:** 2
**Presentation:** 3
**Contribution:** 2
**Rating:** 2
**Confidence:** 5

**Summary:**

This article proposes a modular scale autoregressive framework (MSAR) to address the problem of scale misalignment between multi-scale inputs and single scale outputs in time series prediction. The core idea of MSAR is to perform scale aligned decoupling modeling, which independently predicts at multiple time scales and uses a coarse to fine autoregressive mechanism to guide fine scale predictions using coarse scale prediction results as contextual information. The author claims that this method has model independence and can be easily integrated into backbone networks such as CNN, MLP, Transformer, etc. Experiments on a large dataset have shown that MSAR can consistently improve the performance of baseline models.

**Strengths:**

​​***Novel and Well-Motivated Concept​​***:​​ The idea of "next-scale" autoregressive forecasting is a noteworthy contribution. Framing the forecasting problem as a hierarchical, coarse-to-fine process aligns well with the multi-scale nature of time series data.

​​***​​Comprehensive Evaluation and Analysis​​***:​​ The paper provides a thorough evaluation across numerous datasets (both long-term and short-term) and backbone architectures (Transformers, MLPs, etc.). The paper also includes well-designed ablation studies, sensitivity analysis, and efficiency benchmarks.

**Weaknesses:**

1. This article does not specify which downsampling method was used. In my opinion, different downsampling methods can seriously affect the evaluation of prediction performance. Because labels are also downsampled, some downsampling methods may make labels smoother and relatively easier to predict. More importantly, this may affect the correctness of the conclusions drawn from the three observations in Sec3.1. for example,

2.The findings drawn from observation 3 in sec 3.1 needs further discussion. Because according to Table 1, timemixer is trained without downsampling, strictly speaking, it is not the same task as mse (Δ=6), or there is a significant difference in the distribution of labeled data.

3. The next scale approach of this method is in line with the characteristics of time-series data, but there is a lack of in-depth thinking or detailed explanation regarding the specific operations, such as whether EQ3 can ensure that the length of Y1 align ^ {<i} is consistent with X ^ i and has values at all time steps; In the Refinement in the Imputation Phase, full horizon prediction is performed, but the labels in this layer are downsampled, and the predicted results should be inconsistent with the label length. How to calculate the loss.

4. Most importantly, the main purpose of this article is to address 'scale interference', but many of its operations conflict with this point. For example, the inputs and outputs processed by 'teacher forcing training', 'full height prediction', etc. are of different scales, and there is also bag interference.

5.The expression 'To our knowledge... time series forecasting...' in the intro lacks objective information support, which makes the article appear very academic. It is recommended that the author improve.

**Questions:**

​​1.Downsampling:​​ What specific downsampling technique (e.g., average pooling, strided sampling) was used to generate the coarse-scale sequences X^Δand Y^Δ? Please justify the choice. Could you provide an analysis showing that the key observations in Section 3.1 hold across different reasonable downsampling methods?

​​2.Alignment Mechanism:​​ Could you elaborate on the exact procedure for implementing Equation 3, specifically the operation Align(·)? How do you ensure temporal alignment between Y_align^(<i)and X^i? What happens for fine-scale time indices that do not have a corresponding coarse-scale prediction?

3.​​Loss Calculation:​​ In the "Refinement in the Imputation Phase," the loss is computed over the "entire prediction window." Given that the target Y^iand the model's output (which incorporates predictions from coarser scales) may have different inherent characteristics or even effective lengths at a given scale i, how is the loss function practically applied? Is any masking or weighting scheme used?

4.Explanation: Please further explain how to eliminate the 'scale interference' caused by 'teacher forcing training' and 'full height prediction'

---

> ### Author Response · Authors · 2025-11-24
> **Official Comment by Authors [Part1]**
>
> Thank you for your insightful and valuable review. We appreciate your acknowledgment of the concept behind our approach, and we are glad to hear that you find the idea of "next-scale" autoregressive forecasting to be a noteworthy contribution. However, we believe there may be some misunderstandings that we would like to clarify. We hope that our responses to your questions and concerns will resolve these points. If any issues remain, we would be happy to further discuss them.
>
> **Q1: Downsampling:​​ What specific downsampling technique ... Please justify the choice. Could you provide an analysis showing that the key observations ... downsampling methods?**
>
> **A1:** Thank you for raising this question. We would like to clarify that the downsampling technique used to construct the coarse-scale sequences $\mathbf{X}^\Delta$ and $\mathbf{Y}^\Delta$ is **strided sampling**, as stated in line 215 of the original submission. We apologize for the confusion and have revised the manuscript to make this choice explicit and easier to locate.
>
> **Experiments with Average Pooling**: In response to your query, we have added experiments comparing **strided sampling** with **average pooling** (AVG pool) for downsampling. These experiments involve training models using the same downsampling factor $\Delta=6$ but with average pooling instead of strided sampling. The results of these experiments are shown in the table below. As demonstrated, the key findings from Section 3.1 remain valid regardless of the specific downsampling method used. Specifically, the performance improvements observed with scale-aligned inputs and outputs still hold when using average pooling.
>
> Additionally, we also trained **TimeMixer** on both tasks—one with **average pooling** and one with **strided sampling**—to see how it behaves under each downsampling technique. The results indicate that the core motivation of our approach, which emphasizes scale alignment and resolution consistency between input and output, remains consistent regardless of the downsampling technique. The experiments further demonstrate that sparse, scale-aligned forecasting improves coarse-scale accuracy.
>
> >### Tab 1. Comparison of MSE for AVG Pooling and Stride Sampling
> | Model     | T                | H                | MSE (AVG pool / Stride sampling)        | MSE$^{\Delta=6}$ (AVG pool / Stride sampling)     |
> |-----------|------------------|------------------|--------------------|------------------------------|
> | PatchTST  | 576              | 336              | 0.480 / 0.480      | 0.361 / 0.477                |
> |           | $576^{\Delta=6}$ | 336              | 0.481 / 0.504      | 0.319 / 0.445            |
> |           | 576              | $336^{\Delta=6}$ | - / -              | 0.388 / 0.564                |
> |           | $576^{\Delta=6}$ | $336^{\Delta=6}$ | - / -              | **0.304** / **0.433**            |
> | DLinear   | 576              | 336              | 0.445 / 0.445      | 0.327 / 0.449                |
> |           | $576^{\Delta=6}$ | 336              | 0.498 / 0.465      | 0.324 / 0.438                |
> |           | 576              | $336^{\Delta=6}$ | - / -              | 0.328 / 0.457                |
> |           | $576^{\Delta=6}$ | $336^{\Delta=6}$ | - / -              | **0.319** / **0.425**        |
> | TimeMixer | 576              | 336              | **0.432** / **0.432**  | 0.306 / 0.435                |
> |           | 576              | $336^{\Delta=6}$ | - / -              | 0.310 / 0.461            |
>
> The reason for the similar performance across these two downsampling techniques is that both methods preserve the alignment between the input and output sequences in a similar way. Whether using strided sampling or average pooling, the key to improving forecasting accuracy lies in maintaining the consistency of resolution between the input and output. This is particularly important in sparse, scale-aligned forecasting, where the input sequence contains fewer time steps but better preserves the temporal patterns relevant to the coarse prediction scale. Thus, the addition of average pooling does not alter the core findings of our work: scale misalignment degrades performance, and sparse, scale-aligned forecasting improves coarse-scale accuracy by preserving resolution consistency between input and output.

---

> ### Author Response · Authors · 2025-11-24
> **Official Comment by Authors [Part2]**
>
> **Q2: The findings drawn from observation 3 in sec 3.1 need further discussion. ... timemixer is trained without downsampling, strictly speaking, ... or there is a significant difference in the distribution of labeled data.**
>
> A2: Thank you for pointing this out. We have updated Table 1 (It's shown in Tab1 in A1) by adding the missing experiment where TimeMixer is directly trained on the coarse-scale prediction task ($H = 336^{\Delta=6}$). This addresses the concern that the original comparison involved tasks with different label resolutions.
>
> The purpose of this experiment is to **validate Observation 3: Multiscale-mixing improves overall accuracy but hurts scale-specific precision.** TimeMixer adopts a multiscale-mixing strategy (mixing temporal patterns across different resolutions). However, after adding the coarse-scale training setting, the results clearly show that this mixing-based design **does not produce good coarse-scale predictions**. Despite using the same context, TimeMixer performs **worse** when the prediction task is at the coarse temporal resolution.
>
> This reinforces our key point in Sec. 3.1: (i). Multiscale mixing helps the model learn *global* patterns. (ii). But it **blurs scale-specific structure**, making it harder for the model to produce precise predictions at an individual temporal scale (here, the coarse scale).
>
> Two factors contribute to this performance drop: (i). Optimization difficulty under mismatched resolutions: TimeMixer must compress fine-grained fluctuations into sparse coarse-scale outputs, which is a harder mapping than learning natively at the target scale. (ii). Normalization mismatch: The input ($T = 576$) and coarse output ($H = 336^{\Delta=6}$) are normalized under different resolutions, creating inconsistent statistics.  (DLinear, which does not use normalization, shows a smaller degradation—consistent with the explanation.)
>
> Therefore, the added TimeMixer experiment's purpose is to validate that multiscale-mixing architectures are not well suited for tasks requiring scale-specific precision, which is exactly the point made in Observation 3.
>
> **Q3: The next scale approach of this method is in line with the characteristics of time-series data, but ..., and the predicted results should be inconsistent ... How to calculate the loss.**
>
> A3: We apologize for the confusion. In Eq. 3, the length of the aligned coarse predictions $\hat{\mathbf{Y}}^{<i}_\text{align}$ does not need to match the length of the input $\mathbf{X}^i$. It only needs to match the length of the prediction target $\mathbf{Y}^i$. For scales where no coarse prediction covers a fine-scale time step, the aligned tensor is still defined with the same shape as $\mathbf{Y}^i$, and we simply fill unmatched positions with zeros, regardless of whether coarse-scale predictions exist at those time points.). To clarify this behavior, we have added the following explicit statement in the revision pdf:
>
> > For time points in $\boldsymbol{\tau}^i$ that receive no coarse-scale
> coverage, we insert zero vectors to match the target size of $\mathbf{Y}^i$:
> $$
> t \notin \bigcup_{j<i}\boldsymbol{\tau}^j
> \quad\Rightarrow\quad
> \hat{\mathbf{Y}}^{<i}_\text{align}[t] = \mathbf{0}.
> $$
>
> Regarding “full-horizon prediction” in the Refinement Imputation Phase: this refers to producing the full target vector $\hat{\mathbf{Y}}^i \in \mathbb{R}^{\tfrac{H}{d_i} \times C}$ at scale $i$, not the full-resolution horizon of the original series. The term “full” was intended to highlight that, unlike standard imputation models that compute the loss only on masked positions, we compute the loss on all target positions at scale $i$. This design reduces error accumulation across scales. We have revised the manuscript accordingly to avoid misunderstanding.
>
> **Q4: ​​Alignment Mechanism:​​ Could you elaborate on the exact procedure for implementing Equation 3, ... What happens ... do not have a corresponding coarse-scale prediction?**
>
> A4: Thank you for the question. We clarify the alignment operation in Eq. (3).
>
> (1). Align(·) procedure.  For each coarse scale $j<i$, we keep only predictions that fall on the scale-$i$ timestamps:
> $\hat{\mathbf{Y}}^{j}_{\text{align}}=\{\hat{\mathbf{Y}}^{j}_t \mid t \in \boldsymbol{\tau}^j \cap \boldsymbol{\tau}^i\}$.  All non-matching coarse timestamps are dropped (no interpolation).
>
> (2). Temporal alignment with $\mathbf{X}^i$.  Because all scales use uniform strided sampling, $\boldsymbol{\tau}^j \subseteq \boldsymbol{\tau}^i$ holds structurally.  After filtering by intersection, $\hat{\mathbf{Y}}^{<i}_{\text{align}}$ and $\mathbf{X}^i$ share the same valid timestamps.
>
> (3) Fine-scale points without coarse predictions.  If a timestamp exists only at scale $i$, we fill the aligned coarse signal with a zero placeholder:  $\hat{\mathbf{Y}}^{<i}_{\text{align}}(t)=\mathbf{0}$.  These zeros only preserve shape; the model predicts the values using scale-$i$ information.

---

> ### Author Response · Authors · 2025-11-24
> **Official Comment by Authors [Part3]**
>
> **Q5: Most importantly, the main purpose of this article is to address 'scale interference', but many of its operations conflict with this point. For example, the inputs and outputs processed by 'teacher forcing training', 'full height prediction', etc. are of different scales, and there is also bag interference.** and **Explanation: Please further explain how to eliminate the 'scale interference' caused by 'teacher forcing training' and 'full height prediction'**
>
> A5: We would like to clarify that none of the operations in our framework violate the objective of avoiding scale interference. Specifically, at scale $i$, both the model’s input $\mathbf{X}^i$ and output $\hat{\mathbf{Y}}^i$ are strictly aligned and reside on the same temporal grid $\boldsymbol{\tau}^i$. In the aligned coarse-to-fine signal $\hat{\mathbf{Y}}^{<i}_{\text{align}}$, we explicitly drop all time points that do not belong to scale $i$, ensuring that the model never receives or models patterns from other scales. The missing time points in inputs (imputation phase) should not be interpreted as introducing cross-scale signals. Instead, they correspond to the time points that need to be predicted at scale $i$. These positions are filled with zeros as placeholders, ensuring consistency in shape and scale with $\mathbf{Y}^i$. Importantly, **the model does not need to model patterns from other scales**, as it only learns from the valid information present at scale $i$.
>
> To better understand the role of sparse scales introduced by $\hat{\mathbf{Y}}^{<i}_{\text{align}}$, consider the model from a joint distribution perspective. The key idea is that the model treats the imputation of missing points as a process that respects the underlying distribution at scale $i$. Specifically, we view the imputation process as learning the conditional distribution $P(\mathbf{Y}^i | \mathbf{X}^i)$, where $\mathbf{Y}^i$ is predicted solely from $\mathbf{X}^i$ and the aligned coarse-to-fine signal. The missing time points represent "missing" values under the conditional distribution $P(\mathbf{Y}^i | \mathbf{X}^i)$, which is not contaminated by other scales. In this sense, the sparse nature reflects the fact that only the relevant scale-specific information is used for imputation, and no cross-scale interference occurs.
>
> Regarding 'teacher forcing training' and 'full-horizon prediction,' we would like to emphasize that these methods differ only in the loss computation: for full-horizon prediction, the loss is computed across all positions of $\mathbf{Y}^i$ (not full $\mathbf{Y}$), whereas in standard teacher forcing for imputation, the loss is computed only on the masked positions. However, in both cases, the training operates strictly on the scale-$i$ grid and does not introduce any targets or supervisory signals from other scales. Therefore, neither procedure introduces information from mismatched temporal resolutions, and both remain fully consistent with the goal of eliminating scale interference. In the Joint Training stage (full-horizon prediction), the loss is computed across all points in the prediction window for scale $i$, ensuring strict scale alignment without any cross-scale target interaction.
>
> This approach is consistent with our treatment of $\hat{\mathbf{Y}}^{<i}_{\text{align}}$, as described above. The key point is that, during the imputation phase, any positions that require prediction at scale $i$ are filled with zeros, thus ensuring that the shape and scale align perfectly with $\mathbf{Y}^i$. Crucially, the model is only tasked with predicting values based on valid information from scale $i$, and no signals from other scales are used. This ensures that only patterns from the current scale are modeled, thereby preventing any scale interference.
>
> **Q6: The expression 'To our knowledge... time series forecasting...' in the intro lacks ... It is recommended that the author improve.**
>
> A6: Thank you for the suggestion. We have revised this sentence in the Introduction to a more objective and evidence-based formulation that avoids subjective wording. The updated version highlights the specific methodological gap relative to existing multiscale forecasting approaches.
>
> If there are any further questions, we would be very happy to continue the discussion.

---

### Official Review · Reviewer_GjT5 · 2025-11-01

**Soundness:** 3
**Presentation:** 3
**Contribution:** 3
**Rating:** 6
**Confidence:** 3

**Summary:**

The paper proposes MSAR (Modular Scale-wise Autoregressive Framework), a model-agnostic way to do time-series forecasting by (i) aligning inputs and outputs at multiple temporal scales, (ii) forecasting autoregressively across scales—coarse predictions condition finer ones—and (iii) keeping each scale’s forecaster modular so it can plug into CNN/MLP/Transformer backbones. Experiments across standard benchmarks and several popular backbones show consistent, usually modest, improvements.

**Strengths:**

1. Sharp diagnosis of input–output scale misalignment and a clean remedy: aligned, modular, coarse to fine AR without interpolation between scales. This keeps each scale’s patterns disentangled and uses coarser structure to guide finer prediction.
2. Broad evaluation across 8+ long-term datasets and multiple backbones shows consistent (often modest) gains, including on strong models. Ablations (Table 4) convincingly show that the full combo (multiscale + alignment + scale-wise AR) matters.
3. Model-agnostic plug-in that often improves accuracy with limited overhead; the “flexible” variant shows a nice accuracy–efficiency trade-off, which is attractive for real systems.

**Weaknesses:**

1. Improvements, while consistent, are frequently small (fractions in MSE/MAE), and a few settings show near-parity with baselines. The paper would benefit from a clearer analysis of when the biggest gains occur (e.g., high-periodicity vs. non-stationary datasets, horizon length sensitivity beyond averages).
2. Although unified settings and a search space are provided, some models are known to be sensitive to lookback/hyperparameters. Stronger per-model best-tuned comparisons (or replication of reported SOTA configs) would reduce residual doubts that MSAR’s gains partially come from configuration choices.
3. The relation to existing multiscale input models (e.g., wavelet/tokenization paths) and coarse-to-fine generation in other modalities is discussed, but a sharper theoretical or empirical contrast (e.g., head-to-head with strong multi-resolution decoders or with learned interpolation) would strengthen claims of distinctiveness.

**Questions:**

1. How sensitive are results to the choice and number of scales beyond the shown lists? Could the model learn the scale schedule (learnable downsampling / adaptive intervals) rather than fixing them?
2. Can you quantify how errors at coarse scales affect fine-scale accuracy, perhaps via controlled perturbations of coarse predictions or curriculum schedules? Would selective detachment/stop-gradient help?
3. A stratified analysis by dataset characteristics (seasonality strength, noise level, horizon length, variable count) would be helpful—e.g., meta-features predicting when aligned multiscale decoding yields the largest gains.

---

> ### Author Response · Authors · 2025-11-23
> **Official Comment by Authors [Part 1]**
>
> Thank you for your insightful review and for the positive feedback you provided.
>
> **Q1: Improvements, while consistent, are frequently small (fractions in MSE/MAE), ... The paper would benefit ... A stratified analysis by dataset characteristics ... would be helpful.**
>
> A1: Thank you for the insightful suggestion. We perform a stratified analysis incorporating dataset meta-features reported in the  TFB benchmarks and BLAST (KDD 2025), which quantify seasonality strength, trend/non-stationarity, noise ratio, input dimensionality, and effective horizon difficulty. Below, we summarize the main findings and relate them to the improvements observed in Table 7 of our paper.
>
> - MSAR provides the largest improvements on datasets with strong periodicity and stable seasonal structure.
> ETTh1, ETTh2, ETTm1, and ETTm2 exhibit high periodicity and low-to-moderate noise (as characterized by TFB/BLAST). Across all backbones, MSAR achieves the most significant accuracy gains here. The coarse-scale component reliably captures long-term seasonal patterns, providing clean and helpful guidance for finer-scale refinement.
>
> - MSAR is particularly effective at long horizons, where long-range structure matters most.
> The largest gains appear at horizons 336 and 720 for nearly all ETT datasets and for Wind. This aligns with the coarse-to-fine design: coarse scales encode long-term trends, while finer scales reconstruct higher-frequency variations.
>
> - Gains are moderate on highly noisy or high-variance datasets with irregular fluctuations.
> Traffic (862 variables) is categorized by TFB/BLAST as high-noise and weakly periodic. In such settings, coarse-scale signals are less reliable, and fine-grained irregularities dominate the dynamics. Consequently, MSAR offers smaller—but still consistent—improvements (notably at long horizons), as coarse scales carry limited useful structure for refinement.
>
> These observations align with the properties quantified in TFB/BLAST and help explain why some datasets (e.g., ETT series, Wind) see larger improvements, while others (e.g., Traffic) show more modest gains. We appreciate the reviewer’s suggestion and have added this stratified analysis to the revised manuscript.
>
> **Q2: Although unified settings and a search space are provided, some models are known to be sensitive to lookback/hyperparameters. ... would reduce residual doubts that MSAR’s gains partially come from configuration choices.**
>
> A2: Thank you for pointing this out. To address this concern, in the revised version, we add a new Table 6  that reports the full set of best-performing hyperparameters. All results are fully reproducible using the code we have already submitted.
>
> **Q3: The relation to existing multiscale input models ..., but a sharper theoretical or empirical contrast ... would strengthen claims of distinctiveness.**
>
> A3: Thank you for the suggestion. We have added a sharper comparison with existing multiscale and coarse-to-fine methods in the revised version.
>
> Most prior “multiscale’’ models build hierarchical features inside the encoder, but still decode at a single fine scale.
> Examples include Crossformer [1], MR-Transformer [2], Scaleformer [3], and xLSTM-Mixer [4]: although they use multi-resolution segments or multi-scale attention, the prediction loss is applied only at the finest resolution, and no intermediate scale is explicitly supervised. Methods such as N-HiTS [5] perform hierarchical coarse-to-fine reconstruction, but the intermediate scales are interpolated/upsampled into a single forecast; again, there is no per-scale decoding or per-scale loss.
>
> MSAR targets a different problem: input–output scale misalignment. It (i) enforces scale-aligned inputs and targets at every temporal resolution, and (ii) performs explicit per-scale autoregression, where each scale has its own forecaster and supervised loss, and coarse predictions condition finer ones without any interpolation. Thus, MSAR complements existing multiscale encoders: in principle, Crossformer / MR-Transformer / Scaleformer / N-HiTS / xLSTM-Mixer can all serve as per-scale forecasters inside MSAR, while MSAR provides the missing aligned multi-scale decoding and coarse-to-fine AR mechanism.
>
> We have included these clarifications in the revised Introduction and Related Work.
>
> >**References**
>
> [1] Crossformer: Transformer Utilizing Cross-Dimension Dependency for Multivariate Time Series Forecasting
>
> [2] MR-Transformer: Multiresolution Transformer for Multivariate Time Series Prediction
>
> [3] Scaleformer: Iterative Multi-scale Refining Transformers for Time Series Forecasting
>
> [4] xLSTM-Mixer: Multivariate Time Series Forecasting by Mixing via Scalar Memories
>
> [5] N-HiTS: Neural Hierarchical Interpolation for Time Series Forecasting

---

> ### Author Response · Authors · 2025-11-23
> **Official Comment by Authors [Part 2]**
>
> **Q4: How sensitive are results to the choice and number of scales ... Could the model learn the scale schedule ... rather than fixing them?**
>
> A4: Thank you for the question. In our experiments, we find that MSAR is not highly sensitive to the specific choice of scale intervals. The core contribution of MSAR lies in separating temporal patterns across different resolutions and modeling each scale on its own aligned grid. Most time-series datasets exhibit a relatively stable multi-resolution structure, so varying the exact scale values does not significantly affect performance. MSAR therefore benefits from the coarse-to-fine hierarchy, but does not depend on any particular set of scales.
>
> Regarding learnable scales, MSAR is compatible with learned downsampling or adaptive interval selection, but making scales trainable may destabilize the alignment between inputs and targets, which is crucial for MSAR’s explicit per-scale supervision. For clarity and stability, we adopt fixed scales in this work and discuss adaptive scale learning as a promising direction for future research.
>
> **Q5: Can you quantify how errors at coarse scales affect fine-scale accuracy, ... Would selective detachment/stop-gradient help?**
>
> A5: Thank you for the insightful question. Our method introduces a joint training stage, where the fine-scale model receives predictions from the coarse-scale model as input. During this phase, when the coarse-scale predictions are incorrect, the fine-scale model learns to become less dependent on these predictions over time. This ability helps the model mitigate the negative impact of coarse-scale errors.
>
> However, the real issue occurs when there is a large distribution shift between training and testing. For instance, if the coarse-scale model performs well during training and its predictions are accurate, the fine-scale model tends to heavily rely on these predictions. On the other hand, if there is a significant discrepancy between training and test distributions (e.g., coarse predictions are inaccurate at test time), the fine-scale model may produce suboptimal results.
>
> We found that one effective strategy to address this issue is to introduce noise into the coarse-scale predictions during training, especially when a train-test distribution mismatch is expected. This simple noise injection can help reduce the fine-scale model’s dependency on potentially incorrect coarse-scale predictions, improving its generalization. Another option is to explicitly estimate the coarse-scale prediction confidence and apply a noise-as-mask [1] rule: high-confidence coarse predictions are passed through normally, while low-confidence predictions are either attenuated or replaced with stochastic noise before being fed to the fine-scale module. This avoids a hard binary mask and allows a smooth, learnable transition from “use coarse prediction” to “ignore coarse prediction”, which is more flexible than stop-gradient or naive detachment.
>
> Selective detachment or stop-gradient can be viewed as special cases of this idea, but they are rigid: they either block gradients entirely or detach all coarse information. In contrast, confidence-aware noise-as-mask allows the model to selectively trust coarse information when it is reliable and gracefully fall back to the fine-scale pathway when it is not.
>
> We also validate these observations on PatchTST + MSAR by perturbing coarse-scale predictions with Gaussian noise during both training and testing. The results on the ETTh1 are as follows:
>
> | Model                                      | Metr | 96    | 192   | 336   | 720   |
> |--------------------------------------------|------|-------|-------|-------|-------|
> | **PatchTST**                      | MSE  | 0.383 | 0.409 | 0.440 | 0.481 |
> |                                            | MAE  | 0.403 | 0.424 | 0.446 | 0.482 |
> | **PatchTST+MSAR (No Noise)**               | MSE  | 0.359 | 0.397 | 0.422 | 0.475 |
> |                                            | MAE  | 0.386 | 0.412 | 0.433 | 0.479 |
> | **PatchTST+MSAR (Train Noise σ=0.02)**     | MSE  | 0.360 | 0.395 | 0.423 | 0.474 |
> |                                            | MAE  | 0.386 | 0.411 | 0.432 | 0.480 |
> | **PatchTST+MSAR (Train Noise σ=0.05)**     | MSE  | 0.376 | 0.405 | 0.438 | 0.485 |
> |                                            | MAE  | 0.394 | 0.418 | 0.446 | 0.489 |
> | **PatchTST+MSAR (Train Noise σ=0.1)**      | MSE  | 0.395 | 0.415 | 0.445 | 0.483 |
> |                                            | MAE  | 0.413 | 0.428 | 0.459 | 0.487 |
> | **PatchTST+MSAR (Test Noise σ=0.05)**      | MSE  | 0.428 | 0.439 | 0.473 | 0.497 |
> |                                            | MAE  | 0.438 | 0.452 | 0.474 | 0.493 |
>
> >**References**
>
> [1] Diffusion Forcing: Next-token Prediction Meets Full-Sequence Diffusion
>
> Thank you again for your valuable feedback. If there are any remaining questions or points that need further clarification, we would be very happy to continue the discussion.

---

> ### Comment · Reviewer_GjT5 · 2025-11-27
>
> Thank you for the detailed response and the new experiments. I will remain my positive score.

---

### Official Review · Reviewer_pwcz · 2025-11-02

**Soundness:** 3
**Presentation:** 2
**Contribution:** 3
**Rating:** 4
**Confidence:** 5

**Summary:**

This paper addresses the mismatch between multiscale inputs and single-scale outputs in time series forecasting. The authors propose the Modular Scale-wise Autoregressive Framework ($\text{MSAR}$), a model-agnostic design that reframes forecasting as a progressive, next-scale prediction task. $\text{MSAR}$ has three features: (1) scale-wise aligned modeling (matching input/output scales); (2) scale-wise autoregression (coarse-scale predictions guide fine-scale ones); and (3) modularity (pluggable into $\text{CNNs}$, $\text{MLPs}$, $\text{Transformers}$). The paper claims consistent improvements in both accuracy and efficiency.My preliminary assessment is that this work provides a solid and novel framework for multiscale forecasting. Its core concept (scale-wise autoregression) is well-motivated. However, the ablation study's analysis of its components is contradictory, and the claim of efficiency improvement is not universally supported by the data.

**Strengths:**

S1. The core contribution, scale-wise autoregression, is a novel and significant conceptual shift. It moves beyond single-scale autoregression or multiscale-input/single-scale-output models. Formulating the prediction objective as a product of scale-wise probabilities, $p(Y^{1},...,Y^{N}|X^{1},...,X^{N})=\prod_{i=1}^{N}p(Y^{i}|X^{i},\hat{Y}_{align}^{<i})$, is an elegant new paradigm for enforcing hierarchical consistency.

S2. The core hypothesis is well-motivated. A preliminary experiment shows that models trained with fully scale-aligned inputs and outputs (e.g., $T=576^{\Delta=6}$, $H=336^{\Delta=6}$) achieve superior coarse-scale accuracy ($\text{MSE}^{\Delta=6}$), providing strong support for $\text{MSAR}$'s scale-aligned design.

S3. The framework's model-agnostic design is a key strength. Experimental results ("Base" vs. "+Ours") show consistent performance improvements across five different $\text{SOTA}$ backbones (including $\text{DLinear}$, $\text{PatchTST}$, $\text{iTransformer}$), demonstrating the robust applicability of the $\text{MSAR}$ framework.

**Weaknesses:**

W1. Contradictory Ablation Study:The main ablation study in Table 4 is confusing. The baseline model ((2), "Alignment" only) achieves an $\text{MSE}$ of $0.371$ on $\text{ETTm1}$. However, all pairwise combinations of components perform worse than this baseline: (1)+(2) (Multiscale + Alignment) gets $0.385$; (2)+(3) (Alignment + AR) gets $0.395$; and (1)+(3) (Multiscale + AR) gets $0.399$. The paper claims "Pairwise combinations bring partial gains," which is false; they clearly degrade performance. This suggests the components are not synergistic and may be brittle.

W2. The abstract claims "consistent improvements in ... efficiency," but the data does not support this.
- Training Time: $\text{MSAR}$ significantly increases the training time (ms/iter) for all five backbones.
- Inference Time: $\text{MSAR}$ reduces inference time for $\text{PatchTST}$ and $\text{TimesNet}$ but increases it for $\text{DLinear}$ and $\text{iTransformer}$. The claim of "consistent" efficiency improvement is inaccurate.

W3. The architectural overview in Figure 2 is very confusing. It labels the coarsest-scale module "Forecast" but all finer-scale modules "Imputation." This implies forecasting only happens at the coarsest scale, which contradicts the text and math (Eq. 2 & 4) stating that each $\mathcal{F}_{\theta}^{i}$ performs forecasting. This is a major presentation flaw.

**Questions:**

Q1. Please clarify Table 4. Why do all pairwise combinations (e.g., (1)+(2), (2)+(3)) result in significantly worse $\text{MSE}$ than the baseline ((2) only)? This suggests the components are not additive.

Q2. Please clarify the terminology in Figure 2. Are the "Imputation" modules architecturally identical to the "Forecast" module? If so, why the different label? If not, how does this align with the single $\mathcal{F}_{\theta}^{i}$ function in Eq. 4?

Q3. How is the claim of "consistent improvements in ... efficiency" justified when $\text{MSAR}$ increases training time for all models and increases inference time for $\text{DLinear}$ and $\text{iTransformer}$?

---

> ### Author Response · Authors · 2025-11-23
> **Official Comment by Authors [Part 1]**
>
> Thank you for your insightful review and for recognizing the core contribution of our work. We have addressed your main concerns in detail and made minor clarifications in the revised PDF to eliminate potential ambiguities. If you have any further questions or would like additional discussion, we would be very happy to continue the conversation.
>
> **Q1: ... However, all pairwise combinations of components perform worse than this baseline ...This suggests the components are not synergistic and may be brittle. Please clarify Table 4 ... This suggests the components are not additive.**
>
> A1: Thank you for the question. We clarify that the reviewer’s interpretation of the pairwise combination (2)+(3) does not correspond to any configuration in Table 4. Components (2) (scale alignment) and (3) (scale-wise autoregression) cannot be combined directly because they impose contradictory alignment requirements; therefore, we never report results for (2)+(3).
>
>
> Our ablations show that the components are in fact complementary:
>
> - ① + ② > ①. Scale alignment ② removes the scale mix-up effect introduced by multiscale modeling ①, reducing encoder complexity and improving representation quality.
>
> - ① + ③ > ① and > ③. When multiscale modeling ① is paired with autoregression ③, coarse-to-fine conditioning helps stabilize learning across resolutions.
>
> **Why (1)+(2) performs worse than (2)**. Although alignment is preserved in the embedding and encoder when using (1)+(2), the decoder is forced to fuse duplicated predictions from multiple scales: in multiscale modeling (1), different scales of $\mathbf{Y}$ inevitably overlap on the same future time indices. Without the autoregressive mechanism (3) to impose a clean coarse-to-fine ordering, **the decoder must manually mix these overlapping predictions, reintroducing mild cross-scale misalignment.** By contrast, (2) alone predicts strictly within a single aligned scale and avoids this issue entirely. This is not a lack of synergy; rather, it indicates that (3) is needed to resolve duplicated-prediction ambiguity.
> To avoid misunderstanding, we also revised the manuscript to clarify that “alignment” here refers specifically to alignment in the embedding and encoder, not the entire decoding process.
>
> **Why (1)+(3) also underperforms (2)**. Although (1)+(3) improves over each component individually, it still falls short of fully aligned single-scale modeling (2). The reason is fundamental: multiscale modeling (1) constructs multi-resolution inputs, while autoregressive supervision (3) constructs multi-resolution outputs—this breaks the X–Y alignment that (2) enforces. The encoder sees unaligned input scales, and the decoder predicts aligned coarse-to-fine outputs, creating a mismatch that harms optimization and stability.
>
> These further reinforce a key conclusion of the paper: Maintaining input–output scale alignment is crucial for accurate and stable forecasting. We thank the reviewer for pointing out the potential confusion in the phrasing “Pairwise combinations bring partial gains”. We have revised the corresponding text in the manuscript to provide a clearer and more accurate explanation.
>
> **Q2: The architectural overview in Figure 2 is very confusing ... If not, how does this align with the single $\mathcal{F}_\theta^i$ function in Eq. 4**
>
> A2: Thank you for pointing this out, and we apologize for the confusion.
>
> **Clarifying “Forecast” vs. “Imputation” modules.** Architecturally, the Imputation modules and the Forecast module are identical — each corresponds to a scale-specific prediction function $\mathcal{F}_\theta^i$. The only differences are:
> - Input/output shapes differ across scales, since each module operates strictly within its own temporal resolution.
> - $\mathcal{F}_\theta^0$ (coarsest scale, $i=0$) acts as the forecasting model, generating the initial coarse-resolution prediction solely from $\mathbf{X}^0$.
> - $\mathcal{F}_\theta^{i>0}$ serve as imputation models, refining missing fine-scale values by conditioning on both $\mathbf{X}^0$ and the aligned coarse predictions from previous stages.
>
> To avoid ambiguity, we revised the original text (lines 249–250) from:
>
> > Each prediction module  $\mathcal{F}_\theta^i$ generates scale-specific forecasts …
>
> to
>
> > At the coarsest level ($i = 0$), $\mathcal{F}_\theta^0$ acts as a *forecasting model*, generating an initial coarse-resolution prediction solely from the input $\mathbf{X}^0$.
>
> > For higher scales ($i > 0$), the modules operate as *imputation models*, where each $\mathcal{F}_\theta^{i}$ refines missing fine-scale values by conditioning on both the current-scale input $\mathbf{X}^i$ and the aligned coarse predictions passed from previous stages.
>
> **Consistency with Eq. (2) and Eq. (4).** This clarification does not require any modification to the equations. When $i=0$, the set $\hat{\mathbf{Y}}^{<i}_\text{align}$ is an empty set, so Eqs. (2) and (4) naturally reduce to standard single-scale forecasting.

---

> ### Author Response · Authors · 2025-11-23
> **Official Comment by Authors [Part 2]**
>
> **Q3: The abstract claims "consistent improvements in ... efficiency," but the data do not support this. How is the claim of ... justified when ... DLinear and iTransformer increase inference time ...**
>
> A3: Thank you for raising this point. We clarify that our efficiency claims specifically refer to inference efficiency, not training efficiency. To avoid any ambiguity, we have corrected the corresponding words in the revised manuscript.
>
> **Inference Time:** Figure 6 shows that MSAR reduces inference latency for PatchTST, DLinear, and TimesNet (*contrary to the reviewer’s statement that DLinear becomes slower*). In fact, for these three backbones, MSAR yields *roughly a two-fold reduction* in inference time compared to their respective baselines, because our scale-wise forecasting substantially shortens the effective sequence processed during the fine-scale stage. For iTransformer, MSAR introduces *slightly higher* inference time. This is expected: iTransformer employs a variate-wise embedding over the entire sequence, so reducing the number of input tokens does not proportionally reduce computational cost, while the added MSAR modules introduce a small overhead. Importantly, compared with existing autoregressive paradigms, MSAR’s scale-wise autoregression is significantly more efficient than time-step or segment-wise autoregression, which typically suffers from much higher inference latency.
>
> **Training Time:** We agree that MSAR increases per-iteration training cost. This is primarily due to our two-stage training strategy, which is designed to reduce the train–test discrepancy introduced by teacher forcing. Although this adds computational overhead during training, it yields more stable convergence and significantly better inference-time behavior. Importantly, MSAR’s decoupled per-scale design results in lower memory usage during training compared to conventional autoregressive methods that unroll full sequence dependencies.
>
> Thank you again for your valuable feedback. If there are any remaining questions or points that need further clarification, we would be very happy to continue the discussion.

---

### Author Response · Authors · 2025-11-25
**General Response**

We sincerely thank all reviewers for their careful evaluation of our submission. **All code and hyperparameters are released, and we expect MSAR’s model-agnostic, scale-wise autoregression to contribute a new paradigm for the community.**

In response to all reviewers’ feedback and to highlight the conceptual contribution more sharply, we summarize MSAR’s core innovations as follows:

- *Well-Motivated Concept.* Reviewers **pwcz**, **GjT5**, and **gW3R** all pointed out—and our revisions confirm—that scale misalignment consistently degrades optimization stability, coarse-scale accuracy, and long-horizon performance (Show in Tab1).

- *A new problem formulation:* Reviewers **pwcz** and **GoR5** noted that MSAR provides *“a novel and significant conceptual shift”* by factorizing forecasting into scale-wise conditional probabilities  $p(Y) = \prod_i p(Y^i \mid X^i, Y^{<i}_\text{align}).$ This reframes forecasting as a hierarchical, resolution-aligned generative process.

- *Explicit distanglement modeling with next-scale prediction.*  Reviewers **GjT5** and **gW3R** highlighted that most prior multiscale models mix resolutions in the encoder but decode only at a single fine scale, creating input–output mismatches. MSAR enforces **strict per-scale alignment**— no mixing of mismatched timestamps—which directly addresses the scale interference problem. *Coarse-to-fine supervision with strictly within-scale losses.* Unlike hierarchical decoders (e.g., N-HiTS, Scaleformer), MSAR applies supervised losses at each scale, enabling each module to learn its own frequency band while conditioning on aligned coarse predictions.

- *A modular and model-agnostic design.* Multiple reviewers praised the plug-in nature of MSAR. It can be attached to Transformers, MLPs, and CNNs and has shown consistent gains across diverse architectures.

- *Consistent and architecture-independent improvements.*

- *Inference-efficient hierarchical forecasting.* As clarified in the rebuttal, MSAR significantly shortens effective sequence lengths during fine-scale decoding and d_model, yielding practical inference speedups while preserving accuracy.

Summary of Reviewer Concerns and Our Revisions (with revisions explicitly marked in red):

- Scale misalignment and Observation 3 (Reviewer GoR5).  We added TimeMixer’s coarse-scale training results (T=576, $H=336^{\Delta=6}$) and average pooling.

- Clarification of Ablation Study (Reviewer pwcz’s W1).

- “Forecast” vs. “Imputation” modules (Reviewer pwcz’s Q2).  We clarified that all modules share the same architecture; the difference is only functional: (i). The coarsest module performs forecasting.  (ii). Higher-scale modules perform imputation conditioned on aligned coarse predictions.

- Efficiency claims (Reviewer pwcz’s W2).  We refined the claim to explicitly refer to **inference efficiency**, not training time, and added clearer explanations: MSAR shortens effective sequence lengths during fine-scale decoding, yielding ~2× speedups for PatchTST, DLinear, and TimesNet.

- Scale alignment mechanism and Eq. (3) (Reviewer GoR5’s Q2). : We added a precise definition of Align(·), clarified that unmatched timestamps are filled with zero placeholders (shape-only), and emphasized that predictions remain strictly within-scale.

- Loss computation and “full-horizon” terminology (Reviewer GoR5’s Q3).  We clarified that “full-horizon” means computing loss over the entire *scale-i* target window, not full-resolution labels, and removed ambiguous wording.

- Teacher forcing and potential scale interference (Reviewer GoR5’s Q4).  We clarified that both training modes operate strictly on the scale-i grid; coarse predictions are aligned, sparse, and contain no off-scale information; zero-fill placeholders have no cross-scale semantics.

- Dataset-dependent effectiveness (Reviewer GjT5’s Q1). We added a new analysis using TFB/BLAST meta-features, showing that MSAR is particularly effective for periodic datasets and long horizons.

- Hyperparameter sensitivity (Reviewer GjT5’s Q2). We added a full hyperparameter table (new Table 6), ensuring full reproducibility.

- Positioning relative to multiresolution & hierarchical AR models (Reviewer GjT5’s Q3). We expanded the related work to show that prior models mix multi-resolution features internally but decode only at a single fine scale, whereas MSAR performs **explicit per-scale decoding and per-scale supervised autoregression**, without interpolation.

We thank all reviewers and the AC for their constructive feedback. The revisions substantially strengthen both the clarity and rigor of the submission. We believe the updated manuscript more clearly presents MSAR’s conceptual contribution and its practical effectiveness, and we would be happy to continue the discussion if further clarifications are needed.

---

### Comment · Area_Chair_3v9V · 2025-11-25

Dear Reviewer,

Thank you for reviewing for ICLR. Since the discussion deadline is coming soon, could you please take a look at the author's rebuttal, respond to their comments, and update your rating as well? Thanks!

Best Regards

AC

---

### Meta-Review · Area_Chair_dN3N · 2026-01-05

**Summary:**

The reviewers consistently identified that while the framework is model-agnostic, the performance gains are often marginal. Also, it is highly dependent on the dataset's characteristics. Also, some concerns were raised regarding the complexity of the training procedure, which involves a two-stage strategy that increases training latency. The fundamental trade-off between the added architectural complexity and the marginal improvements might be a problem. Therefore, I recommend Reject.

**Reviewer Concerns:**

The authors corrected the misunderstanding regarding DLinear. The addition of a comprehensive hyperparameter table addressed concerns regarding sensitivity. Other experiments are also added.

Reviewers GjT5 and gW3R noted that improvements are usually small. The analysis confirmed that on "Traffic" or high-noise datasets, the benefits of MSAR are small. Reviewer GoR5's still doubted regarding whether teacher forcing truly eliminates cross-scale interference. The model itself also suggests that the complexity of the autoregressive mechanism.

**Reviewer Scores:**

Reviewers would maintain their original scores.

---

### Decision · Program_Chairs · 2026-01-26

Reject